# Climatology of the Linke and Unsworth–Monteith Turbidity Parameters for Greece: Introduction to the Notion of a Typical Atmospheric Turbidity Year

**Harry D. Kambezidis *** and **Basil E. Psiloglou** 

Atmospheric Research Team, Institute of Environmental Research and Sustainable Development, National Observatory of Athens, GR-11810 Athens, Greece; bill@noa.gr
* Correspondence: harry@noa.gr

**Abstract:** Solar rays are attenuated by the Earth's atmosphere. This attenuation can be expressed by the turbidity parameters; two of them are the Linke turbidity factor ($T_L$) and the Unsworth–Monteith turbidity coefficient ($T_{UM}$). In this sudy, both parameters are estimated for 33 sites across Greece, and the notion of a Typical Atmospheric Turbidity Year (TATY) is also introduced. Use of the modified clearness index ($k'_t$) is made, while a suggestion for a modified diffuse fraction ($k'_d$) is given. The adoption of the four climatic zones in Greece for energy purposes is made, where the variation of $T_L$ and $T_{UM}$ is studied during a TATY under all and clear-sky conditions. The analysis shows maximum levels in both parameters in late winter–early spring in morning and evening hours, with minimum values at midday. The intra-annual variation of the parameters shows maximum values around March and August and minimum values in summertime and late winter. Maps of annual mean $T_L$ and $T_{UM}$ values over Greece show persistent minimum values over Peloponnese and maximum values over South Ionian Sea. Linear expressions of $T_{UM}$ vs. $T_L$ are derived for all sites under all and clear-sky conditions. Finally, linear expressions for $k'_d$ vs. $k'_t$ are given for all sites and sky conditions.

**Keywords:** atmospheric turbidity; Linke turbidity factor; Unsworth–Monteith turbidity coefficient; Typical Atmospheric Turbidity Year; modified clearness index; modified diffuse fraction; Greece

---

## 1. Introduction

Solar radiation reaching the surface of the Earth undergoes attenuation due to the absorption and scattering of the solar rays by the atmospheric constituents [1]. Therefore, the solar radiation levels measured on the surface of the Earth depend on the concentration of water vapor, nitrogen dioxide, ozone, mixed gases and atmospheric aerosols. Although empirical–analytical expressions have been provided in the international literature for the atmospheric transmittances of the first four constituents, such expressions would not have been possible for aerosols if the notion of atmospheric turbidity were not introduced. The atmospheric turbidity expresses the attenuation of solar rays by aerosols. Several indices (called atmospheric turbidity parameters) have been introduced for this purpose; these are the Ångström turbidity coefficient ($\beta$) and exponent ($\alpha$) [2–4], the Schüepp turbidity factor ($B$) [5], the Linke turbidity factor ($T_L$) [6,7] and the Unsworth–Monteith turbidity coefficient ($T_{UM}$) [8].

There is a growing interest in atmospheric turbidity worldwide as this issue is related to air-pollution studies in cities, solar radiation modelling, atmospheric chemistry, and climate studies. Apart from the pioneering works defining the various atmospheric turbidity parameters mentioned above, numerous studies have been carried out since the 1970s. A sample of such studies is given here.

Polavarapu [9] attempted to evaluate $T_L$ for an urban, a suburban, a rural and an arctic station in Canada. Abdelrahman et al. [10] reported $T_L$ levels for a desert environment (Dhahran, Saudi Arabia). Grenier et al. [11] developed a model for deriving $T_L$ by means of updated spectral extraterrestrial irradiances and extinction coefficients of gaseous absorbers. Eaton [12] discussed the influence of atmospheric boundary-layer stability on atmospheric turbidity and optical turbulence in a desert environment. Cucumo et al. [13] studied the variation of the Linke turbidity factor at two Italian localities, while Diabate et al. [14] evaluated the same factor for several sites in Africa. Zakey et al. [15] studied the atmospheric turbidity levels at two sites in Egypt. Hove and Manyumbu [16] estimated the Linke turbidity factor over Zimbabwe. Bilbao et al. [17] studied the turbidity coefficients in central Spain, while Saad et al. [18] reported the spatial and temporal variability of atmospheric turbidity over Tunisia. Ulscka-Kawalkowsa et al. [19] studied the Linke turbidity factor over Warsaw and Belsk, Poland. A recent study [20] has investigated the accuracy of three different methodologies (in essence, different data inventories) in assessing high atmospheric-turbidity episodes over north-central Spain. The authors found that all three methodologies have a 60% coincidence in identifying such episodes.

In Greece, studies about atmospheric turbidity have been conducted mostly for Athens [21–27] and Thessaloniki [28]. Most of them refer to the Linke turbidity factor. Therefore, there is a gap in recent turbidity studies for Athens and a complete absence of research for Greece as a whole. This gap is intended to be covered in the present work.

The structure of the paper is as follows. Section 2 presents the methodology for calculating the Linke and Unsworth–Monteith turbidity parameters (Section 2.1), as well as the data collection (Section 2.2) and the data processing (Section 2.3). Sections 3.1 and 3.2 present the month–hour graphs of $T_L$ and $T_{UM}$ for selected sites under all and clear-sky conditions, respectively. Section 3.3 accounts for the effect of the clearness index on $T_L$ and $T_{UM}$, while Section 3.4 refers to the intra-annual variation of $T_L$ and $T_{UM}$ over Greece under all and clear-sky conditions. Section 4 presents the main conclusions of the study.

## 2. Materials and Methods

### 2.1. Methodology

2.1.1. Calculation of the Linke Turbidity Factor, $T_L$

$T_L$ was introduced by Linke [6,7] as a parameter to indicate how many (hypothetical) clean and dry atmospheres are necessary to derive the observed attenuation of the extraterrestrial radiation caused by the real atmosphere under clear-sky conditions. $T_L$ is typically bounded between 1 and 10. According to Linke:

$$B_e = S \cdot G_{e,o} \cdot sin\gamma \cdot exp(-m' \cdot \overline{k_r} \cdot T_L) \tag{1}$$

where $B_e$ is the direct horizontal solar radiation equal to $G_e$ - $D_e$, $S$ is the correction factor for the Sun–Earth distance [29] given by Equation (3), $G_{e,o}$ is the solar constant equal to 1361.10 Wm$^{-2}$ [30], $\gamma$ is the solar elevation, $m'$ is the pressure-corrected optical air mass (Equation (5)), which includes the altitude of the site at which $T_L$ is to be estimated, and $\overline{k_r}$ is the mean attenuation of the direct solar radiation due to Rayleigh scattering alone [31,32]. Solving Equation (1) for $T_L$, we obtain

$$T_L = \frac{lnG_{e,o} - lnB_e + lnS + lnsin\gamma}{\overline{k_r}\, m'} \tag{2}$$

in which

$$S = 1 + 0.033\,cos\left(\frac{2\,\pi\,D}{365}\right) \tag{3}$$

$$\overline{k_r} = 6.5567 + 1.7513\,m' - 0.1202\,m'^{\,2} + 0.0065\,m'^{\,3} - 0.00013\,m'^{\,4} \tag{4}$$

$$m' = \frac{P_z}{P_o}\,m \tag{5}$$

$$m = 1\,/\left[sin\gamma + 0.50572(\gamma + 6.07995)^{-1.6364}\right] \tag{6}$$

The optical air mass, $m$, is given by Kasten and Young [33]. In Equation (5), $P_z$ is the atmospheric pressure (in hPa) at an altitude $z$ (in m), and $P_0$ the atmospheric pressure at sea level. Its value is usually taken as equal to 1000 hPa or 1013.25 hPa; the latter value has been adopted in this work. If $P_z$ is not known, it can be calculated via [34]

$$P_z = P_o \cdot exp\left(-\frac{z}{8435.2}\right). \tag{7}$$

Unfortunately, $T_L$ has the disadvantage of depending on air mass. To overcome this difficulty, various authors have tried to derive air-mass-independent $T_L$ expressions. The most popular method to date is the normalization of the estimated $T_L$ values at $m = 2$ ($T_{L2}$) [11,35]. Linke recognized this problem of the turbidity factor and tried—without success—to introduce a new extinction coefficient based on an atmosphere with a water-vapor content of 1 cm. Nevertheless, the present study does not adopt $T_{L2}$ as this factor refers to a specific range of solar elevation angles equivalent to $m = 2$ throughout a year. For a climatology of $T_L$ over a region, the statistics of $T_L$ should be based on all estimated $T_L$ values that represent the real atmospheric conditions over the area for a year or a number of years. This last statement has been considered in the present work.

2.1.2. Calculation of the Unsworth–Monteith Turbidity Coefficient, $T_{UM}$

Unsworth and Monteith [8] introduced their turbidity coefficient, which expresses the absorption of solar rays by a dust-laden atmosphere relative to a dust-free one, with both atmospheres having a specified water-vapor content. $T_{UM}$ typically varies in the range $0 < T_{UM} \leq 1$. According to Unsworth–Monteith,

$$B_e = B_e^* \cdot sin\gamma \cdot exp(-m' \cdot T_{UM}) \tag{8}$$

from which

$$T_{UM} = \frac{lnB_e^* - lnB_e + lnsin\gamma}{m'} \tag{9}$$

where $B_e^*$ is the direct solar radiation in a dust-free atmosphere with a specified water-vapor content. This quantity is given by Bird and Hulstrom [36,37]:

$$B_e^* = S \cdot G_{e,o} \cdot T_r \cdot T_o \cdot T_g \cdot T_w \tag{10}$$

where the $T_i$ values are the atmospheric transmittances (i = r, o, g, w) due to the Rayleigh scattering ($T_r$), attenuation by the ozone ($T_o$), mixed gases ($T_g$), and water-vapor ($T_w$) content in the atmosphere.

The general transmittance function $T_i$ for seven main atmospheric gases (water vapor, ozone, and mixed gases; i.e., $CO_2$, $CO$, $N_2O$, $CH_4$ and $O_2$), can be expressed by the equation proposed by Psiloglou et al. [38–41]:

$$T_i = 1 - \frac{A_1 \cdot m'u_i}{(1 + A_2 \cdot m'u_i)^{A3} + A_4 \cdot m'u_i} \tag{11}$$

where the $A_i$ values ($i = 1 \ldots 4$) are numerical coefficients that depend on the specific extinction process given in Table 1 as proposed by Psiloglou et al. [38–41]; $u_i$ implies the absorbed amount in a vertical column. More specifically, $u_w$ (in cm) is the content of water vapor, $u_o$ (in atm-cm) is the content of ozone, and $u_i$ (in atm-cm) is the content of mixed gases [39,41].

**Table 1.** Values of the coefficients $A_i$ in Equation (11) [38–41].

| Gas | $A_1$ | $A_2$ | $A_3$ | $A_4$ |
|---|---|---|---|---|
| $H_2O$ | 3.0140 | 119.300 | 0.6440 | 5.8140 |
| $O_3$ | 0.2554 | 6107.26 | 0.2040 | 0.4710 |
| $CO_2$ | 0.7210 | 377.890 | 0.5855 | 3.1709 |
| CO | 0.0062 | 243.670 | 0.4246 | 1.7222 |
| $N_2O$ | 0.0326 | 107.413 | 0.5501 | 0.9093 |
| $CH_4$ | 0.0192 | 166.095 | 0.4221 | 0.7186 |
| $O_2$ | 0.0003 | 476.934 | 0.4892 | 0.1261 |

The broadband transmittance function due to the total absorption by the uniformly mixed gases can then be calculated by [39,41]

$$T_g = T_{CO2} \cdot T_{CO} \cdot T_{N2O} \cdot T_{CH4} \cdot T_{O_2} \tag{12}$$

where the transmittances $T_{CO2}, T_{CO}, T_{N2O}, T_{CH4}$ and $T_{O2}$ are given by Equation (11) using the appropriate coefficients, as proposed by Psiloglou et al. [39].

The transmittance corresponding to the Rayleigh scattering is calculated according to the method of Psiloglou et al. [42]:

$$T_r = exp\left[ -0.1128 \cdot m'^{\,0.8346}\left(0.9341 - m'^{\,0.9868} + 0.9391 \cdot m'\right)\right] \tag{13}$$

For the estimation of the total amount of water vapor in a vertical column (the so-called precipitable water, in cm), the following expression proposed by Leckner [43] was used:

$$u_w = \frac{0.493 \cdot e_m}{t} \tag{14}$$

where $e_m$ is the partial water-vapor pressure (in hPa) given by

$$e_m = \frac{e_s \cdot RH}{100} \tag{15}$$

where $RH$ is the relative humidity at the station's height (in %), and $e_s$ is the saturation water-vapor pressure (in hPa), given by Gueymard [44]:

$$e_s = exp\left(22.329699 - 49.140396 \cdot t_o^{-1} - 10.921853 \cdot t_o^{-2} - 0.39015156 \cdot t_o\right) \tag{16}$$

with $t_o = t/100$ and $t$ being the air temperature at the station's height (in K).

For the estimation of the ozone-column content $u_o$ (in atm-cm or in DU (Dobson Units); 1 DU = 0.001 atm-cm), the adjusted van Heuklon ozone model [45] for the Europe area, based on TOMS (Total Ozone Mapping Spectrometer) data in the period 1975–2005, and proposed by Karavana-Papadimou [46], was used:

$$u_o = 0.260 + \{0.0763 + 0.0489 \cdot \sin\left[0.9865\ (D - 17.85)\right] - 0.00144 \cdot \sin\left[3\ (\lambda + 51.2)\right]\} \cdot \sin^2(1.497\ \varphi) \quad (17)$$

where $\lambda$ and $\varphi$ are the geographical longitude and latitude, respectively, of the location.

### 2.1.3. Classification of the Sky Conditions

Kambezidis [47] gives an account of ten possible ways to identify clear skies from available meteorological and/or solar radiation measurements. Nevertheless, he states that the most-adopted technique is the clearness index, $k_t$, which is defined as the ratio $G_e/G_{e,extra}.\sin\gamma$. As regards this index, Reindl et al. [48] have proposed values of $k_t > 0.6$ and $k_t < 0.2$ for clear and cloudy skies, respectively. Li and Lam [49] and Li et al. [50] used values of $0 < k_t \leq 0.15$, $0.15 < k_t \leq 0.7$ and $k_t > 0.7$ for overcast, intermediate and clear skies, respectively, in Hong Kong. Kuye and Jagtap [51] used $k_t > 0.65$ and $0.12 \leq k_t \leq 0.35$ for very clear and cloudy skies, respectively, to classify the sky conditions at Port Harcourt, Nigeria. As $k_t$ depends on $m$, Perez et al. [52] have defined a modified clearness index, $k'_t$, independent of the air mass:

$$k'_t = \frac{k_t}{0.1 + 1.031 \cdot exp\left[-1.4/\left(0.9 + \frac{9.4}{m}\right)\right]} \quad (18)$$

This work adopts the modified $k_t$ using the following limits: $0 < k'_t \leq 0.3$, $0.3 < k'_t \leq 0.65$, and $0.65 < k'_t \leq 1$ for overcast, intermediate and clear skies, respectively.

Another index defining the sky conditions is the diffuse fraction, $k_d$ (e.g., [53–56]), which is the ratio $D_e/G_e$. This sky-condition parameter also depends on air mass through the solar radiation components. To avoid this dependency, the present work defines a modified diffuse fraction, $k'_d$, for the first time worldwide, following the formulation of $k'_t$ in Equation (18):

$$k'_d = \frac{k_d}{0.1 + 1.031 \cdot exp\left[-1.4/\left(0.9 + \frac{9.4}{m}\right)\right]} \quad (19)$$

Figure 1 shows an example of the variation of $k'_t$ vs. $k'_d$ for the site of Alexandroupoli (see Table 2 for site information). It is seen that the dependence of $k'_t$ on $k'_d$ has an exponential decay shape (Equation (20)) and not a linear one as is the case of the original indices; for example, $k_t = 0.860 - 1.015 k_d$ in Hijazin [54]. It is worth mentioning that both paradigms of Alexandroupoli and Hijazin refer to hourly values of the clearness index and the diffuse fraction. The best-fit curve to the hourly data points of Alexandroupoli in Figure 1 has the following form:

$$k'_t = 1.1430 \cdot exp\left(-1.1030k'_d\right) - 0.4346 \cdot exp\left(-5.5970k'_d\right) \quad (20)$$

Expressions of $k'_t$ vs. $k'_d$ for seven other sites have also been derived (see Section 3.3). Further, the characterization of the sky conditions by using $k'_d$ alone, as done with $k'_t$, is risky, because $k'_d$ depends also upon $D_e$, which is not always measured in the same manner as $G_e$. However, the derivation of a relationship of the form $k'_d = f(k'_t)$ may be useful as it can provide information about $D_e$, if $G_e$ measurements are available in a location.

Figure 2 provides information about the distribution of clear, intermediate and overcast skies over eight sites in Greece following the $k'_t$ limitations. The other sites in Table 2 show similar patterns (not shown here).

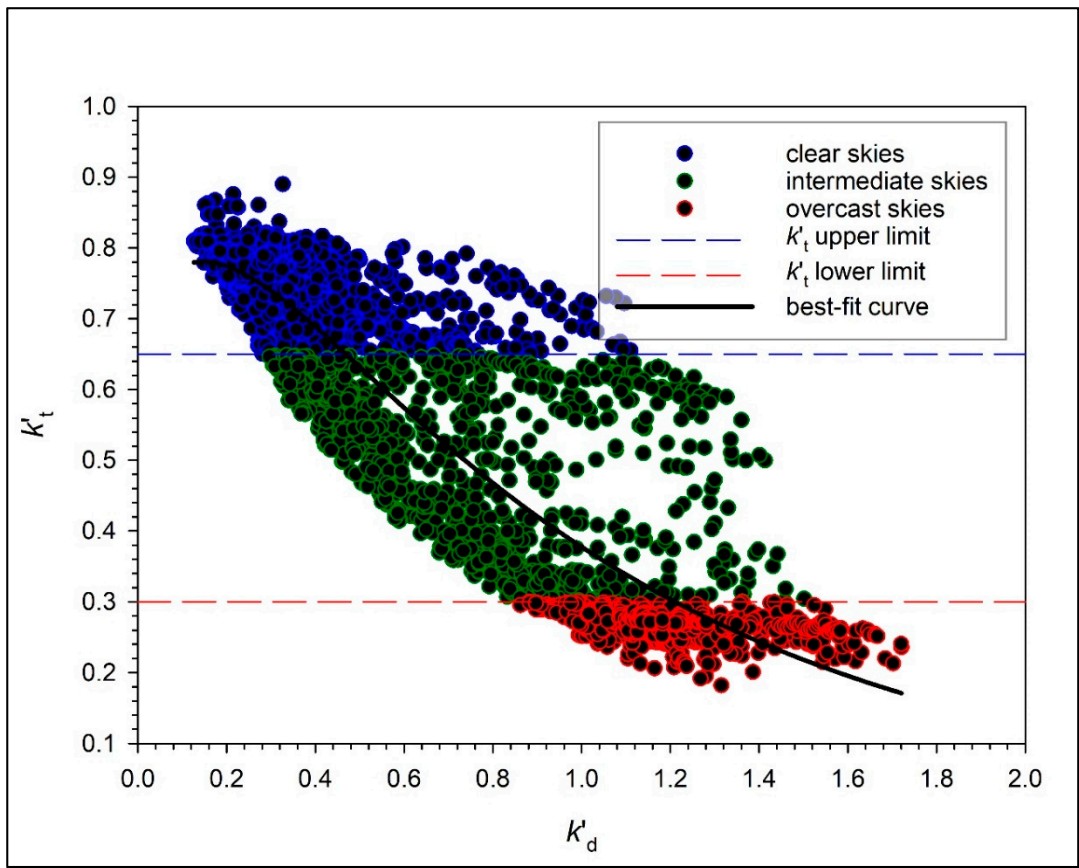

**Figure 1.** Graph of hourly $k'_t$ vs. hourly $k'_d$ values over one complete year for the site of Alexandroupoli; the limits of overcast, intermediate and clear skies are shown. The solid black line is the best-fit curve to the data points from Equation (20) with $R^2 = 0.82$.

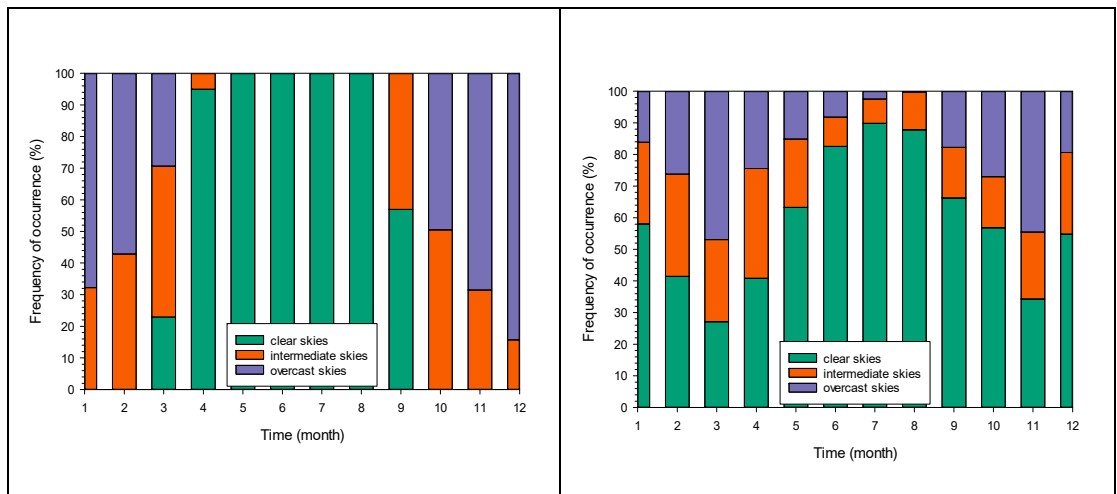

**Figure 2.** *Cont.*

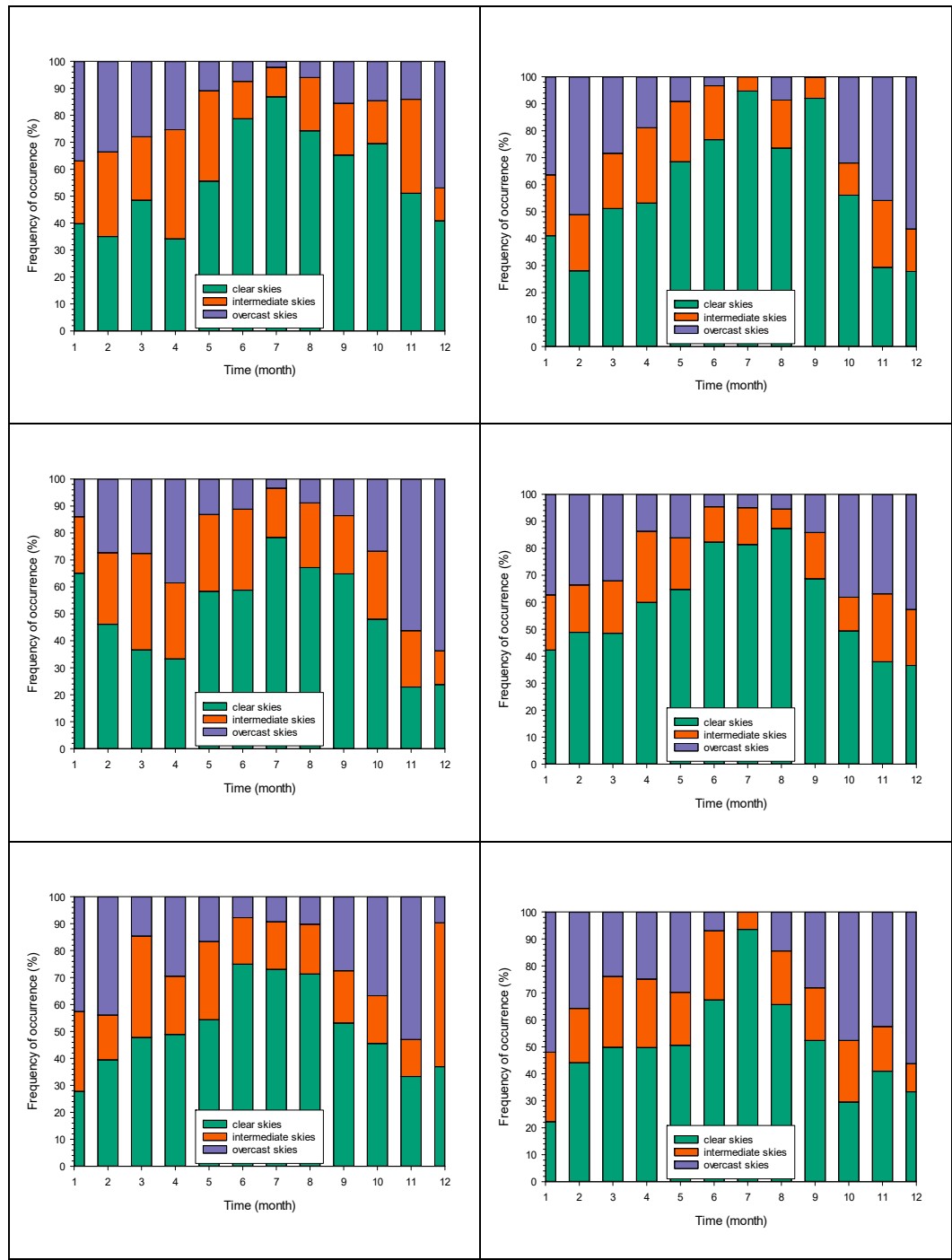

**Figure 2.** Bar charts showing the frequency of occurrence (in %) of the sky conditions over a complete year in climatic zone A: first raw, Irakleio (left), Kalamata (right); climatic zone B: second raw, Agrinio (left), Lesvos (right); climatic zone C: third raw, Alexandroupoli (left), Tripoli (right); climatic zone D: fourth raw: Kastoria (left), Serres (right). For information about the four climatic zones in Greece, see Section 2.2.

*2.2. Data Collection*

Under the auspices of the KRIPIS-THESPIA-II project – funded by the General Secretariat of Research and Technology of Greece, the Atmospheric Research Team of the National Observatory of Athens derived TMYs (Typical Meteorological Years) for 33 locations across Greece. A TMY is a complete year of data consisting of hourly or daily values of meteorological and/or radiometric

parameters; each TMM (Typical Meteorological Month) of the TMY is chosen via a specific statistical procedure for that month, which is closer to the long-term climatic characteristics of the site for that month. The generation of a TMY is implemented from a database containing the selected parameters in a span of years—usually 15–20 and most preferably 30 years. More details about the generation of the TMYs for Greece are given in Kambezidis et al. [57], where data of meteorological parameters were obtained from 33 HNMS (Hellenic National Meteorological Service) stations during the period 1985–2014 (30 years; see Table 2). For each of the 33 sites, five different TMYs were derived for equal applications; i.e., TMY-Meteorology-Climatology (TMY-MC), TMY-Bio-Meteorology (TMY-BM), TMY-Agro-Meteorology-Hydrology (TMY-AMH), TMY-Energy Design for Buildings (TMY-EDB), and TMY-Photovoltaics (TMY-PV). Each type of TMY contains the necessary meteorological and/or solar radiation parameters. The solar radiation ($G_e$, $B_e$, $D_e$) values for the purpose of TMY generation at each of the 33 stations were derived from the available meteorological data (ambient temperature, relative humidity, atmospheric pressure, sunshine duration) via the MRM (Meteorological Radiation Model; see [58–61]), because the HNMS stations do not measure solar radiation. Each of those TMYs consists of hourly values of the parameters considered. For the purpose of this work, the databases of the TMYs-PV at the 33 sites have been selected because the climatology of the atmospheric turbidity is more related to solar radiation/energy applications. Table 2 gives the geographical coordinates, climatic zones and altitudes of the 33 stations, while Figure 3 shows their location on the map of Greece. This map also shows the four climatic zones of the country for energy applications [62]. The delimitation of these zones has been based on three parameters: the heating degree days (HDD), the cooling degree hours (CDH) and the available SSR (surface solar radiation) in a region. Table 3 shows the range of those parameters for each climatic zone.

**Table 2.** List of the meteorological HNMS stations used in this work. WMO: World Meteorological Organization.

| Station | Station Name (Region) | Station's WMO Code (16xxx) | $\varphi$ (Deg. N) | $\lambda$ (Deg. E) | Climatic Zone | $z$ (M amsl) |
|---|---|---|---|---|---|---|
| 1 | Serres (Central Macedonia) | 606 | 41.083 | 23.567 | D | 34.5 |
| 2 | Kastoria (Western Macedonia) | 614 | 40.450 | 21.283 | D | 660.9 |
| 3 | Mikra (outskirts of Thessaloniki, Central Macedonia) | 622 | 40.517 | 22.967 | C | 4.8 |
| 4 | Alexandroupoli (Eastern Macedonia and Thrace) | 627 | 40.850 | 25.933 | C | 3.5 |
| 5 | Kozani (Western Macedonia) | 632 | 40.283 | 21.783 | D | 625.0 |
| 6 | Kerkyra (known as Corfu, Ionian Islands) | 641 | 39.617 | 19.917 | A | 4.0 |
| 7 | Ioannina (Epirus) | 642 | 39.700 | 20.817 | C | 484.0 |
| 8 | Larisa (Thessaly) | 648 | 39.650 | 22.450 | C | 73.6 |
| 9 | Limnos (Northern Aegean) | 650 | 39.917 | 25.233 | B | 4.6 |
| 10 | Anchialos (Thessaly) | 665 | 39.217 | 22.800 | B | 15.3 |
| 11 | Lesvos (Northern Aegean) | 667 | 39.067 | 26.600 | B | 4.8 |

**Table 2.** *Cont.*

| Station | Station Name (Region) | Station's WMO Code (16xxx) | $\varphi$ (Deg. N) | $\lambda$ (Deg. E) | Climatic Zone | $z$ (M amsl) |
|---|---|---|---|---|---|---|
| 12 | Agrinio (Western Greece) | 672 | 38.617 | 21.383 | B | 25.0 |
| 13 | Lamia (Sterea Ellada) | 675 | 38.850 | 22.400 | B | 17.4 |
| 14 | Andravida (Western Greece) | 682 | 37.917 | 21.283 | B | 15.1 |
| 15 | Skyros (Sterea Ellada) | 684 | 38.900 | 24.550 | B | 17.9 |
| 16 | Araxos (Western Greece) | 687 | 38.133 | 21.417 | B | 11.7 |
| 17 | Tanagra (Sterea Ellada) | 699 | 38.317 | 23.550 | A | 139.0 |
| 18 | Chios (Northern Aegean) | 706 | 38.350 | 26.150 | B | 4.0 |
| 19 | Tripoli (Peloponnese) | 710 | 37.533 | 22.400 | C | 652.0 |
| 20 | Elliniko (Attica) | 716 | 37.900 | 23.750 | B | 15.0 |
| 21 | Zakynthos (known as Zante, Ionian Islands) | 719 | 37.783 | 20.900 | A | 7.9 |
| 22 | Samos (Northern Aegean) | 723 | 37.700 | 26.917 | A | 7.3 |
| 23 | Kalamata (Peloponnese) | 726 | 37.067 | 22.000 | A | 11.1 |
| 24 | Naxos (Southern Aegean) | 732 | 37.100 | 25.533 | A | 9.8 |
| 25 | Methoni (Peloponnese) | 734 | 36.833 | 21.700 | A | 52.4 |
| 26 | Spata (Attica) | 741 | 37.967 | 23.917 | B | 67.0 |
| 27 | Kythira (Attica) | 743 | 36.133 | 23.017 | A | 166.8 |
| 28 | Thira (Southern Aegean) | 744 | 36.417 | 25.433 | A | 36.5 |
| 29 | Souda (Crete) | 746 | 35.550 | 24.117 | A | 140.0 |
| 30 | Rodos (known as Rhodes, Southern Aegean) | 749 | 36.400 | 28.117 | A | 11.5 |
| 31 | Irakleio (also written as Heraklion, Crete) | 754 | 35.333 | 25.183 | A | 39.3 |
| 32 | Siteia (Crete) | 757 | 35.120 | 26.100 | A | 115.6 |
| 33 | Kasteli (Crete) | 760 | 35.120 | 25.333 | A | 335.0 |

The names of the stations in the second column are presented as Latin characters, adapted from the Greek alphabet according to the ELOT 743 standard [63], which is based on the ISO 843 [64] standard.

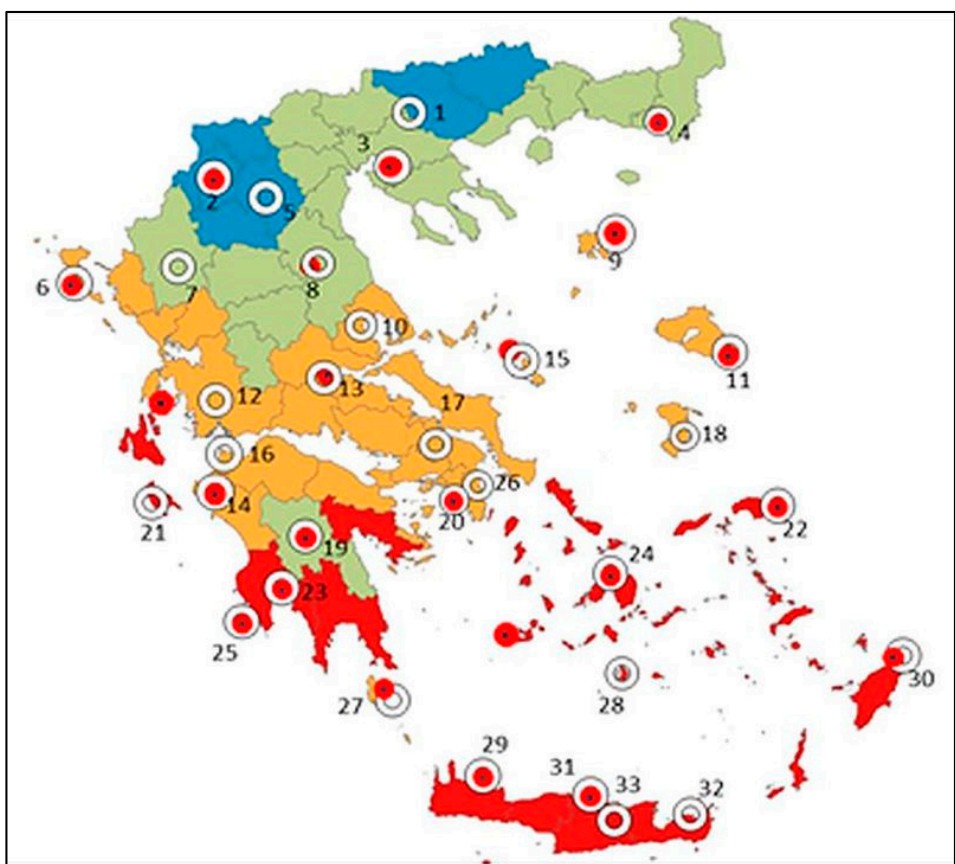

**Figure 3.** Map of Greece showing the location of the 33 Hellenic National Meteorological Service (HNMS) stations (white circles) across the 4 climatic zones: A (red), B (orange), C (green), and D (blue) according to TOTEE (Technical Guide of the Technical Chamber of Greece) [62]. The numbers in the map correspond to those in the first column of Table 2. The red dots correspond to locations in which IWYEC2s (International Weather Years for Energy Calculations) have been derived by ASHRAE (American Society of Heating, Refrigerating and Air-Conditioning Engineers) [65] in the period 1982–1999; they are shown for comparison.

**Table 3.** Range of annual heating degree days (HDD), cooling degree hours (CDH) and available SSR (surface solar radiation) values to define the climatic zones in Greece for energy saving in buildings [61]. The calculation of HDD and CDH considers the following cold and warm periods of the year: (i) climatic zones A, B: 1 November–15 April and 15 May–15 September, respectively, (ii) climatic zones C, D: 15 October–30 April and 1 June–31 August, respectively.

| Climatic Zone | HDD (Dimensionless) | CDH (Dimensionless) | SSR (kWh m$^{-2}$ y$^{-1}$) |
|---|---|---|---|
| A | <1000 | [1300, 4500] | [1700, 1900] |
| B | [1000, 1500] | [2200, 5500] | [1500, 1700] |
| C | [1500, 2000] | [1200, 3800] | [1450, 1600] |
| D | ≥2000 | ≤1500 | ≤1500 |

The difference in handling the sites in this work and that in Kambezidis et al. [57] is that Serres has been classified here in the climatic zone D. The reason is that this site is exactly on the border of two zones (C and D, see map in Figure 3); Kambezidis el. [57] considered it in zone C.

### 2.3. Data Processing

Each TMY-PV at any of the 33 sites in Greece consists of hourly values of ambient temperature (in °C), relative humidity (in %), and global horizontal and direct horizontal solar radiations (in Wm$^{-2}$). As no atmospheric pressure data were included in the TMYs-PV, this parameter was adopted from the corresponding TMYs-MC. All of the above parameters refer to the station's altitude and have gone through quality-control testing during the generation process of the TMYs (see [57]).

The original SUNAE (Sun's Azimuth and Elevation) routine, first introduced by Walraven [66], together with its modifications [67–70] was applied to the geographical coordinates of the 33 stations at every 30 min past the hour for a complete year. To derive the hourly values of the solar altitude, $\gamma$, needed for the calculation of $m$ at every site for a complete year, the SUNAE algorithm was applied to each month of the site's TMY (for description see [57]). Only values of $\gamma$ greater than or equal to 5° were considered (to avoid the cosine effect); thus, the corresponding $T_L$, $T_{UM}$, $k'_t$, and $k'_d$ values were discarded. Another data-quality criterion was the rejection of all $T_L$, $T_{UM}$, $k'_t$, and $k'_d$ values if $G_e$ or $B_e$ were equal to zero. On the other hand, the distinction of skies into overcast, intermediate and clear for the purposes of this work has been based on the criteria mentioned in Section 2.1.3. Furthermore, the hourly values of $T_L$, $T_{UM}$ have been kept in the ranges given in Sections 2.1.1 and 2.1.2, respectively.

The derivation of $T_L$ and $T_{UM}$ at the 33 sites was based on necessary measured or estimated parameters ($B_e$, $P_z$ for $T_L$; $B_e$, $RH$, $T$ for $T_{UM}$); their values have been taken from the available TMY datasets at the 33 locations. Therefore, one could say that the derived $T_L$ and $T_{UM}$ annual datasets also correspond to a TMY; thus, the notion of a TATY (Typical Atmospheric Turbidity Year) is introduced here.

## 3. Results

### 3.1. $T_L$ and $T_{UM}$ Variation: Month–Hour Diagrams for All-Sky Conditions

Kambezidis et al. [71] first introduced the methodology for the month–hour distribution of a meteorological parameter through its application to daylight levels in Athens. Those researchers listed the advantages of using such an analysis. This methodology is also followed in the present study. Due to the large number of sites considered in this work, it is not possible to show diagrams for all 33 locations; therefore, a pair of representative sites per climatic zone has been chosen (as in Figure 2); these are Irakleio and Kalamata for zone A, Agrinio and Lesvos for zone B, Alexandroupoli and Tripoli for zone C, and Kastoria and Serres for zone D.

Figure 4 shows the month–hour graphs of $T_L$ under all-sky conditions for the selected sites. A general observation is the relatively high values of $T_L$ during almost all day in January, February, March or even early April. Lower values are observed at midday. The morning and evening peaks (8 a.m. and 5–6 p.m.) may correspond to rush hours as the meteorological data used to calculate $T_L$ comes from HNMS stations mostly situated at airports and/or close to towns. Similar observations to ours have been reported over individual years in Rome and Arcavacata di Rende, Italy [13], various locations in Zimbabwe [16], and over Sfax in Tunisia [18]. In late spring, autumn and winter, $T_L$ decreases in the evening because of its sensitivity to low solar elevation angles (higher air masses).

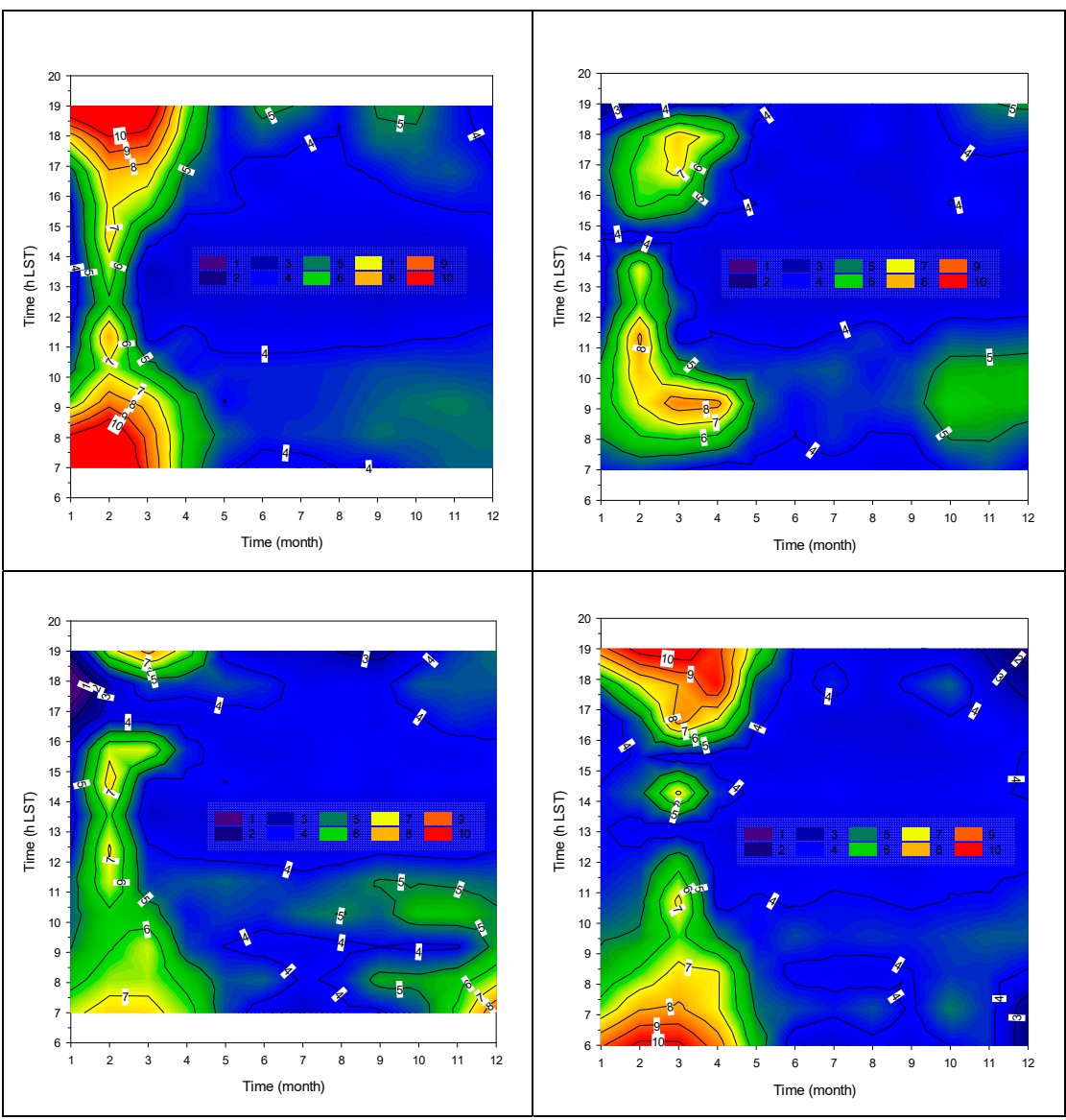

**Figure 4.** *Cont.*

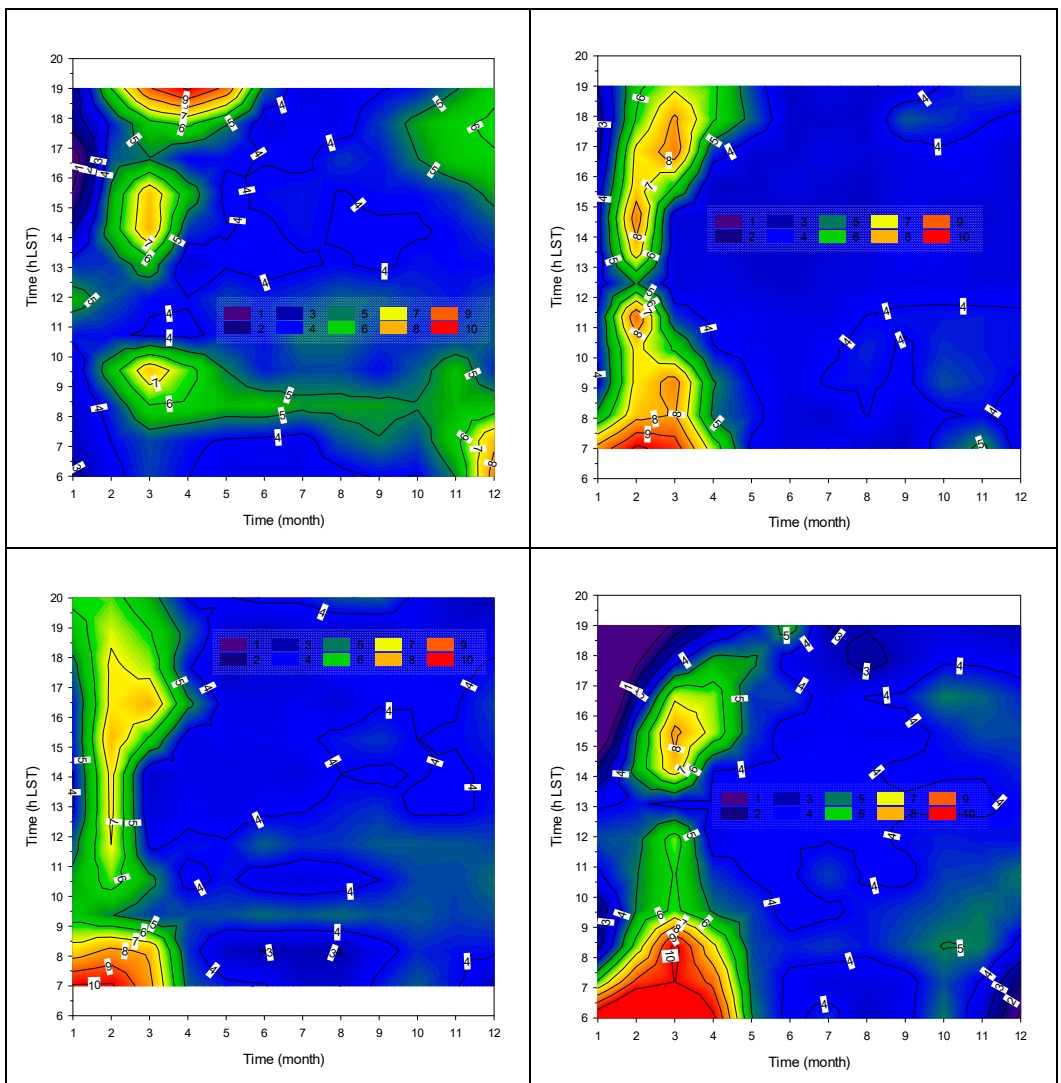

**Figure 4.** Month–hour diagrams of the $T_{\mathrm{L}}$ values in a Typical Atmospheric Turbidity Year (TATY) under all-sky conditions in climatic zone A: first raw, Irakleio (left), Kalamata (right); climatic zone B: second raw, Agrinio (left), Lesvos (right); climatic zone C: third raw, Alexandroupoli (left), Tripoli (right); climatic zone D: fourth raw: Kastoria (left), Serres (right). The color code refers to the $T_{\mathrm{L}}$ levels, which have been kept in the range $1 \le T_{\mathrm{L}} \le 10$. The 06.00–20.00 h LST (Local Standard Time = Universal Time + 2h for Greece) interval limitation in the graphs is due to the adopted criterion $\gamma \ge 5°$.

The observations drawn from Figure 4 may have another explanation. According to the definition of the Linke turbidity factor, during the late autumn–winter period, more clean–dry atmospheres are required to produce the observed attenuation of solar radiation because of extended cloudiness and higher humidity in the atmosphere in relation to those in the spring–early autumn period. Another observation from Figure 4 is that the winter $T_{\mathrm{L}}$ values become lower around noon. This occurs for two further reasons: (i) $T_{\mathrm{L}}$ is inversely proportional to the air mass (or solar elevation angle; see Equation (2)), and it therefore decreases at higher $m'$ values, and (ii) cloudiness and/or ambient humidity are reduced at midday. Similar patterns have been found at the other 25 sites (not shown here).

Figure 5 is equivalent to Figure 4, but it refers to $T_{\mathrm{UM}}$. $T_{\mathrm{UM}}$ also depends on $1/m'$ as $T_{\mathrm{L}}$, but it has an additional disadvantage that the dust-laden and dust-free atmospheres (see Equation (9)) contain a specified water-vapor content; i.e., the content that exists in the atmosphere at the time of observation ($u_w$ in Equation (14)). This differentiates its behavior slightly in relation to that of $T_{\mathrm{L}}$, as $T_{\mathrm{L}}$ refers to a dry atmosphere only. Nevertheless, the $T_{\mathrm{L}}$ and $T_{\mathrm{UM}}$ patterns for the same site look

very similar; one might observe that, if the $T_L$ values are divided by 10, the $T_{UM}$ values are obtained. Kambezidis et al. [24] have come to a similar conclusion. The other 25 sites present very similar $T_{UM}$ patterns (not shown here).

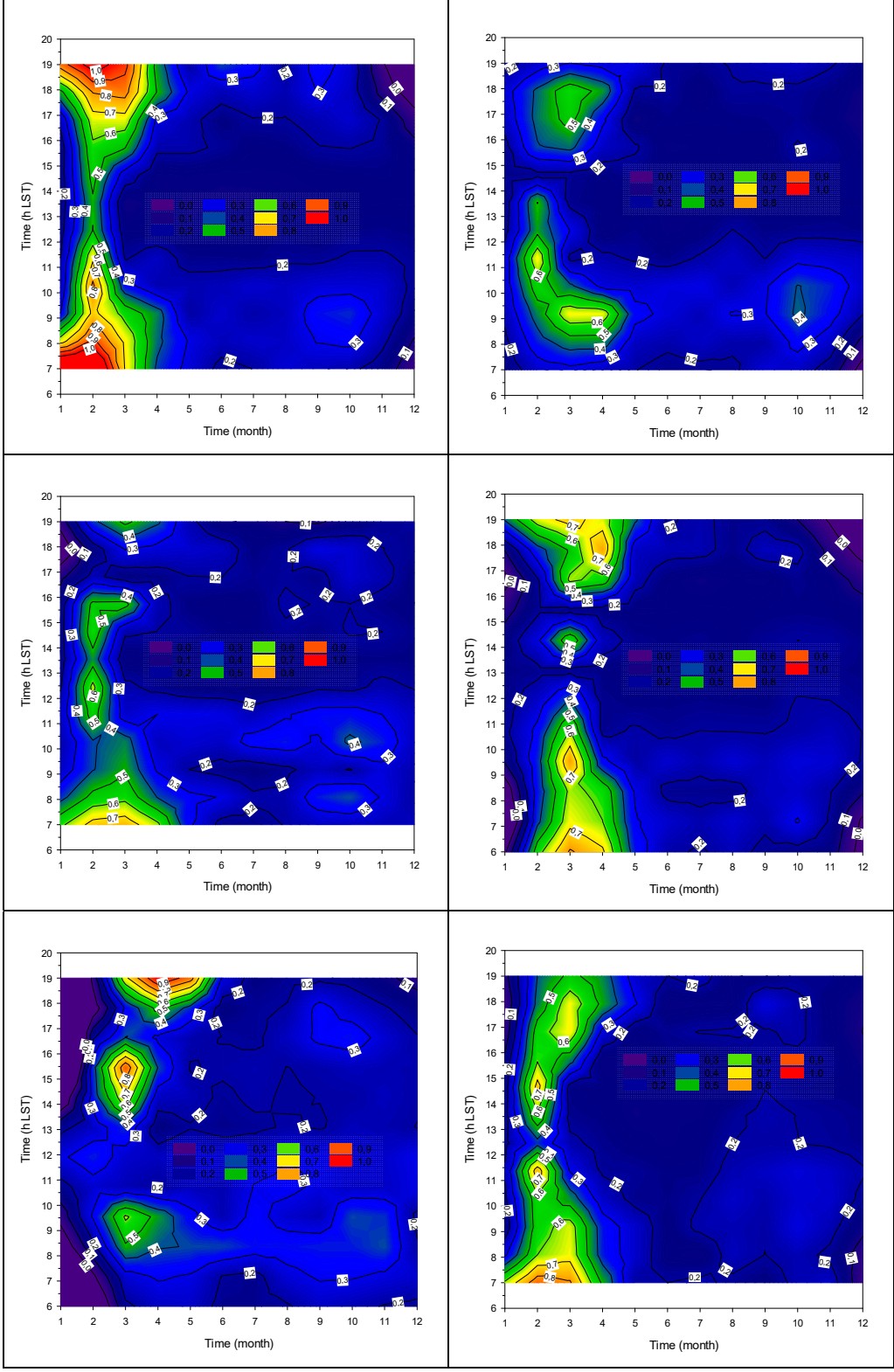

**Figure 5.** *Cont*.

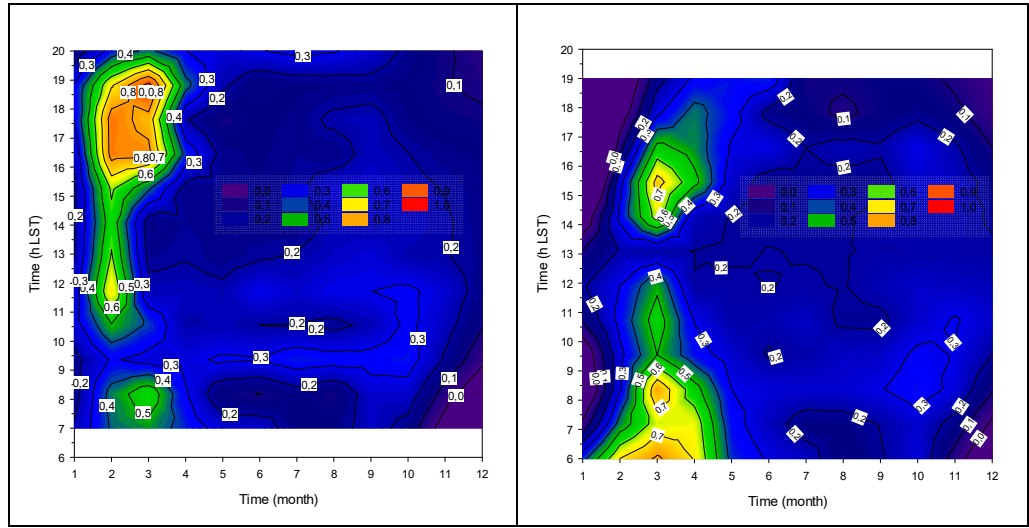

**Figure 5.** Month–hour diagrams of the $T_{UM}$ values in TATY under all-sky conditions in climatic zone A: first raw, Irakleio (left), Kalamata (right); climatic zone B: second raw, Agrinio (left), Lesvos (right); climatic zone C: third raw, Alexandroupoli (left), Tripoli (right); climatic zone D: fourth raw: Kastoria (left), Serres (right). The color code refers to the $T_{UM}$ levels, which have been kept in the range $0 < T_{UM} \leq 1$. The 06.00–20.00 h LST interval limitation in the graphs is due to the adopted criterion $\gamma \geq 5°$.

## 3.2. $T_L$ and $T_{UM}$ Variation: Month–Hour Diagrams for Clear-Sky Conditions

Figure 6 shows the month–hour graphs of $T_L$ under clear-sky conditions for the selected sites in each climatic zone. In these diagrams, the all-sky $T_L$ patterns seem to be repeated in a clearer way. The morning and evening peaks are retained in the period January–March due to the rush-hour activities. Maximum $T_L$ values are also found in the morning hours throughout the year but are lower than those in late winter–early spring; this occurs because of progressively reduced values of relative humidity. In the evening (late spring–early winter), the $T_L$ values are lower than the morning values for the same reason given for $T_L$ under all-sky conditions. A similar pattern exists for the other 25 sites (not shown here).

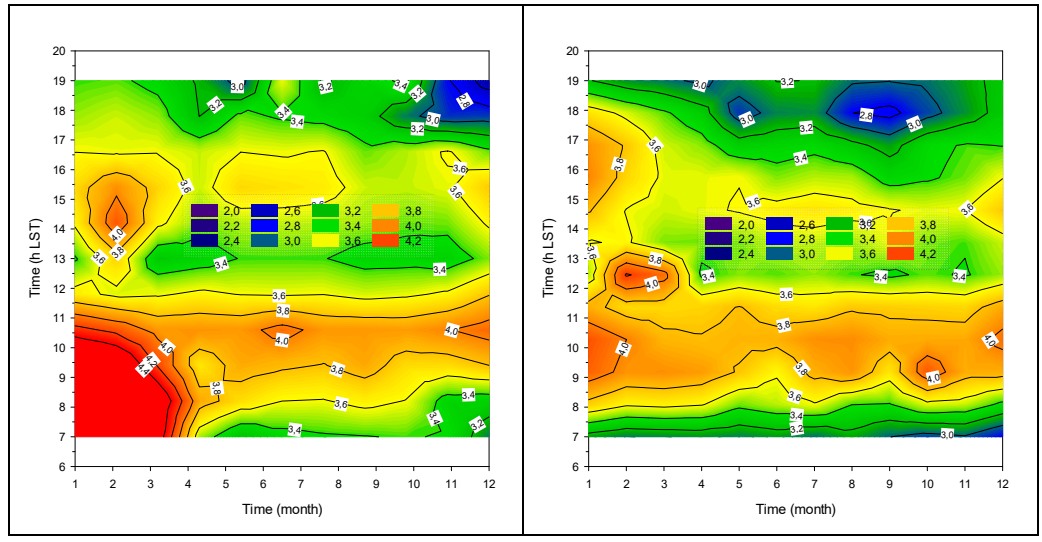

**Figure 6.** *Cont.*

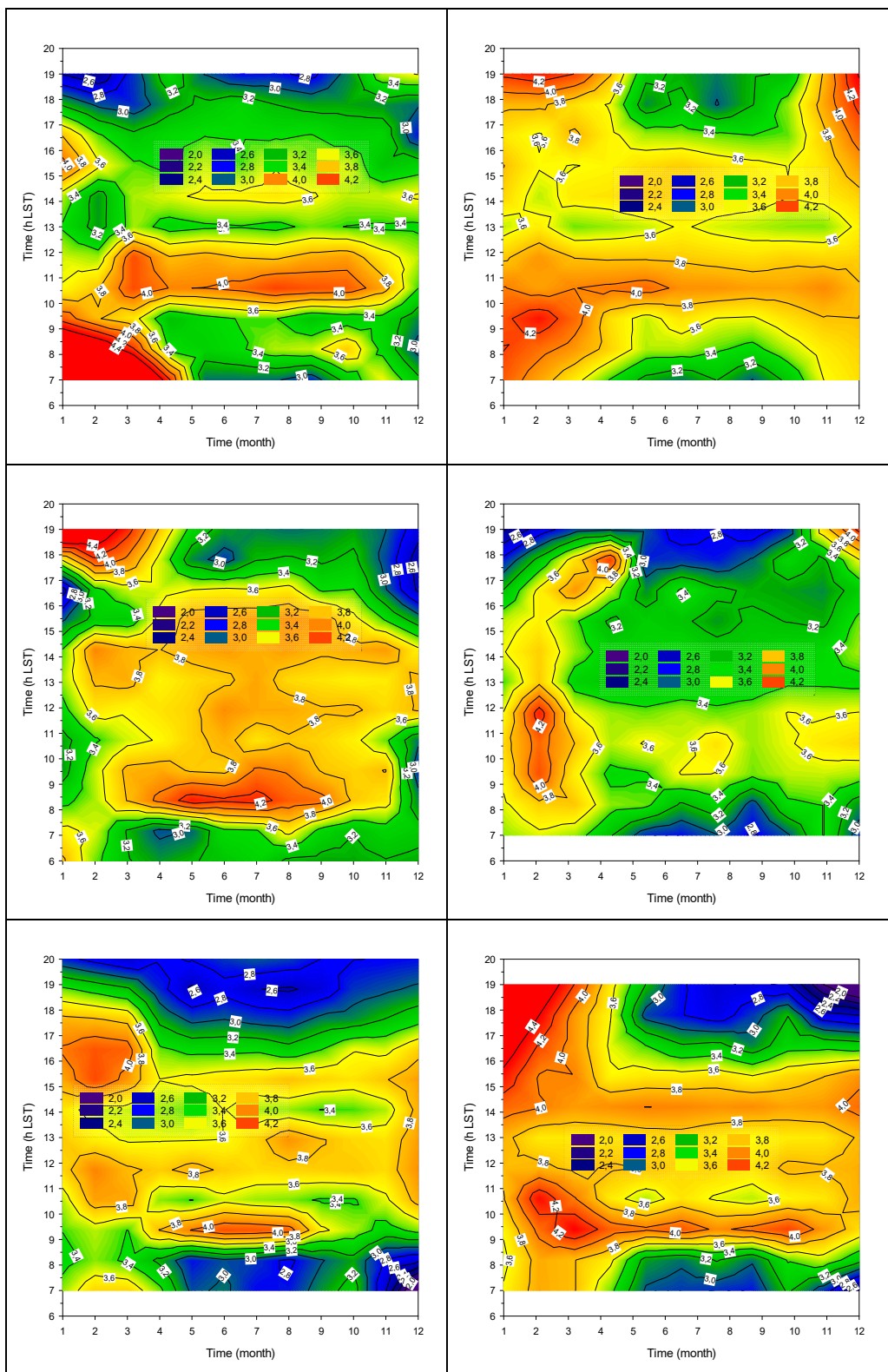

**Figure 6.** Month–hour diagrams of the $T_L$ values in TATY under clear-sky conditions in climatic zone A: first raw, Irakleio (left), Kalamata (right); climatic zone B: second raw, Agrinio (left), Lesvos (right); climatic zone C: third raw, Alexandroupoli (left), Tripoli (right); climatic zone D: fourth raw: Kastoria (left), Serres (right). The color code refers to the $T_L$ levels, which have been kept in the range $2 \le T_L \le 4.2$. The 06.00–20.00 h LST interval limitation of the graphs is due to the adopted criterion $\gamma \ge 5°$.

The $T_{UM}$ pattern for clear skies (Figure 7) follows that of $T_L$ as in the case of all-sky conditions. The explanation is the same. Dividing the $T_L$ levels by 10 for any of the eight sites, one almost obtains the $T_{UM}$ levels for this site, in agreement with Kambezidis et al. [24] for Athens in the period 1975–1984. Similar $T_{UM}$ patterns for the other 25 sites have been obtained (not shown here).

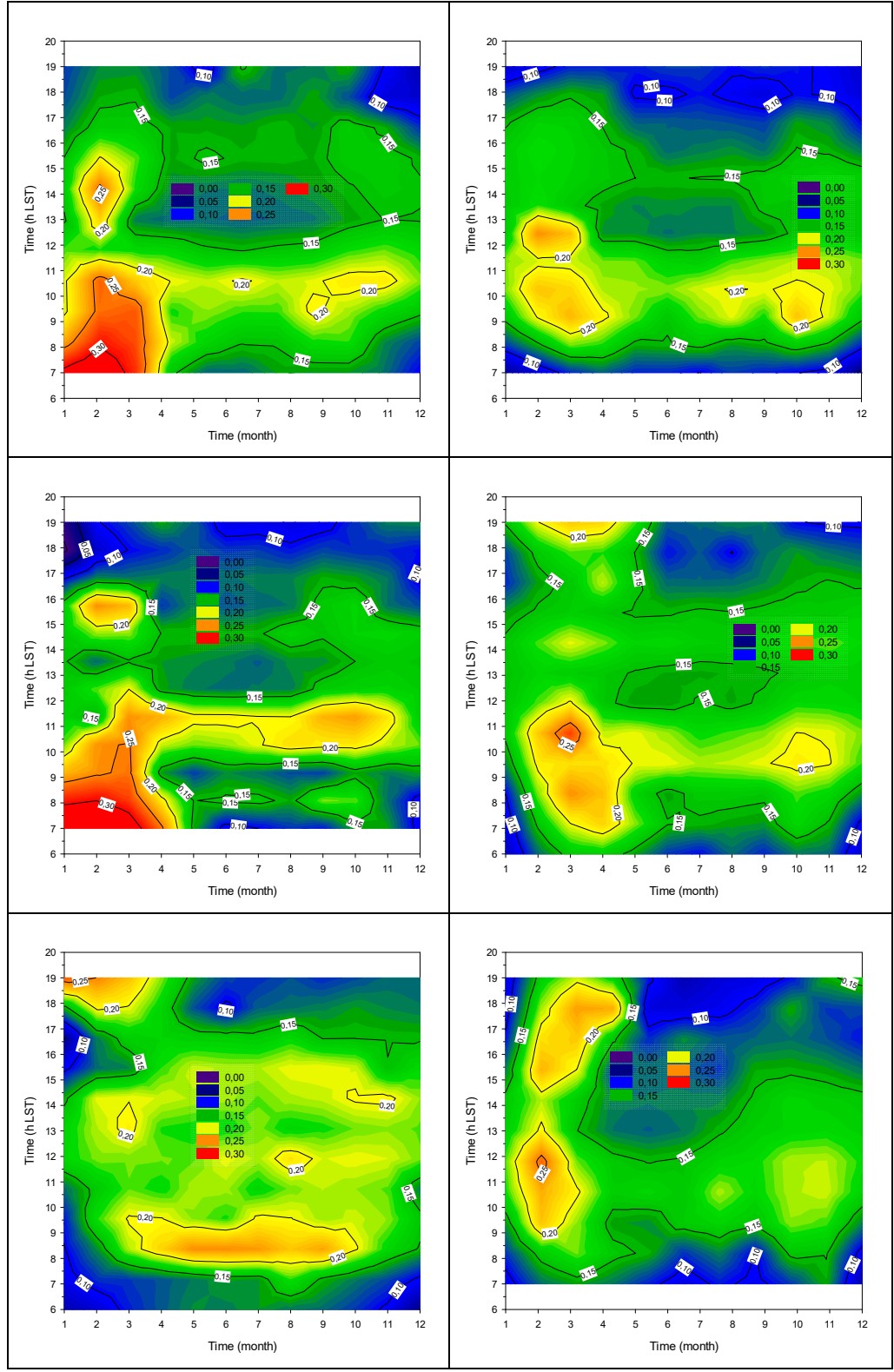

**Figure 7.** *Cont.*

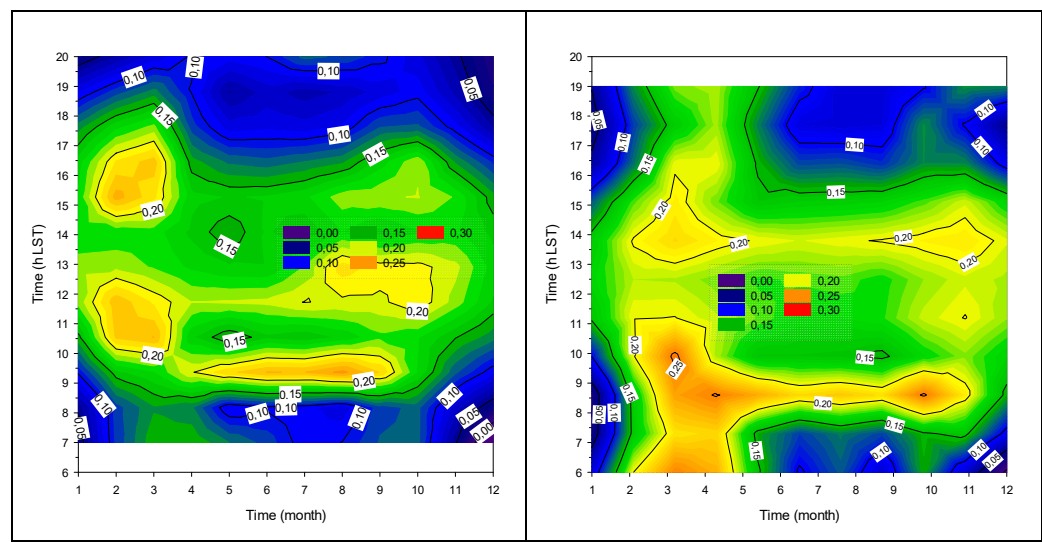

**Figure 7.** Month-hour diagrams of the $T_{UM}$ values in TATY under clear-sky conditions in climatic zone A: first raw, Irakleio (left), Kalamata (right); climatic zone B: second raw, Agrinio (left), Lesvos (right); climatic zone C: third raw, Alexandroupoli (left), Tripoli (right); climatic zone D: fourth raw: Kastoria (left), Serres (right). The color code refers to the $T_{UM}$ levels, which have been kept in the range $0 < T_{UM} \leq 0.3$. The 06.00–20.00 h LST interval limitation of the graphs is due to the adopted criterion $\gamma \geq 5°$.

### 3.3. Variation of $k'_d$ vs. $k'_t$ over Greece

An expression of $k'_d$ as function of $k'_t$ is very useful as $D_e$ can be derived from $k'_d$ provided there are available $G_e$ measurements in a location, as already mentioned in Section 2.1.3. As shown in Figure 1, the hourly values provide a wide dispersion due to extreme weather effects. In order to smooth out any such unwanted effect on the $k'_d$ vs. $k'_t$ relationship, monthly mean values of these indices were considered for the eight selected sites, following the philosophy of the eight representative sites in the four climatic zones of Greece. On the other hand, monthly values are sufficient for engineering applications. Table 4 provides the coefficients for the linear $k'_d$ vs. $k'_t$ equations under all-sky conditions. On the other hand, by grouping the monthly mean $k'_d$ vs. $k'_t$ data pairs of the stations which belong to the same climatic zone, new best-fit lines were obtained and are shown in Figure 8. It is seen from the linear regression equations that the best-fit lines of the zonal pairs A, B, C, and D are very close to each other. This gives the freedom to derive an equation for all climatic zones (Figure 9).

**Table 4.** Coefficients and $R^2$ of the relationship $k'_d = a + b\, k'_t$ for the eight selected sites in Greece. The relationships have been derived from monthly mean $k'_d$, $k'_t$ values under all-sky conditions during TATY.

| Climatic Zone | Site | a | b | $R^2$ |
|:---:|:---:|:---:|:---:|:---:|
| B | Agrinio | 1.8616 | −2.1402 | 0.93 |
| C | Alexandroupoli | 1.7406 | −1.9311 | 0.92 |
| A | Irakleio | 1.7201 | −1.9172 | 0.99 |
| A | Kalamata | 1.6518 | −1.7932 | 0.91 |
| D | Kastoria | 2.0177 | −2.3781 | 0.94 |
| B | Lesvos | 1.7920 | −1.9125 | 0.98 |
| D | Serres | 1.8999 | −2.1945 | 0.94 |
| C | Tripoli | 1.8737 | −2.1123 | 0.96 |

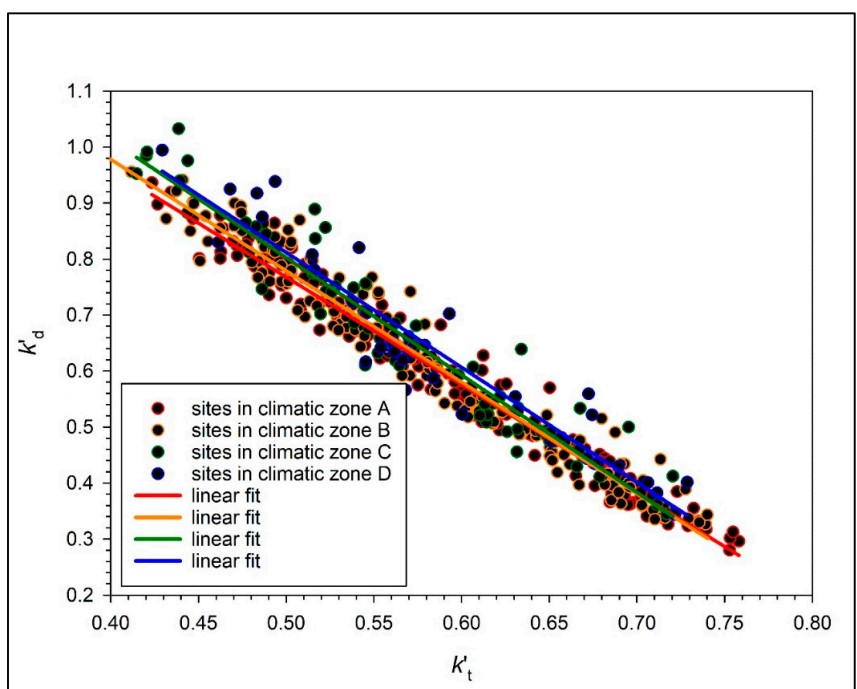

**Figure 8.** Monthly mean $k'_d$ vs. $k'_t$ data pairs for sites belonging to the same climatic zone under all-sky conditions in TATY and their best-fit lines. Linear regression equations for (i) climatic zone A: $k'_d$ = 1.730−1.925 $k'_t$, $R^2$ = 0.97; (ii) climatic zone B: $k'_d$ = 1.774−1.989 $k'_t$, $R^2$ = 0.96; (iii) climatic zone C: $k'_d$ = 1.851−2.097 $k'_t$, $R^2$ = 0.92; (iv) climatic zone D: $k'_d$ = 1.839−2.054 $k'_t$, $R^2$ = 0.90. The lower $R^2$ values in the climatic zones C and D—and especially in D—are expected because of the fewer data points and harsher weather conditions at these sites in comparison to those in the other two climatic regions.

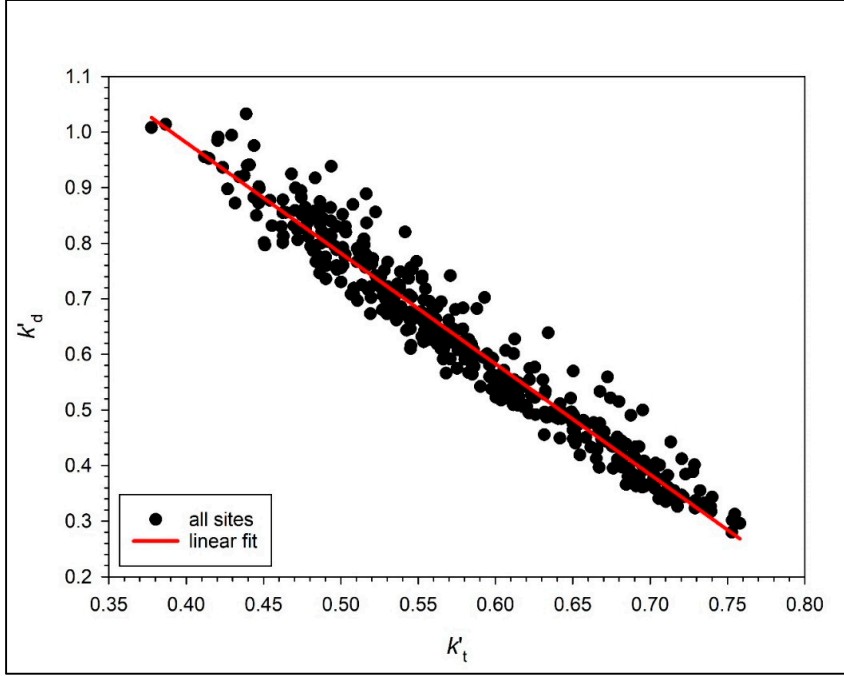

**Figure 9.** Monthly mean $k'_d$ vs. $k'_t$ data pairs for all 33 sites belonging to the four climatic zones under all-sky conditions in TATY and their best-fit line. Linear regression equation: $k'_d$ = 1.777−1.9991 $k'_t$, $R^2$ = 0.94.

A corresponding regression analysis of the $k'_d$ vs. $k'_t$ data for clear skies has been carried out for the eight selected sites (not shown here) giving linear regression equations with $R^2$ much less than 0.40. This could be anticipated since $D_e$ included in $k'_d$ can vary more than $G_e$ during clear weather depending on the aerosol loading in the atmosphere; on the contrary, during all-sky conditions, $D_e$ is reduced and co-varies with $G_e$.

### 3.4. Intra-Annual Variation of $T_L$ and $T_{UM}$

Figure 10 presents the variation of the monthly mean $T_L$ values at all 33 stations under all and clear-sky conditions. The average and $\pm 1\sigma$ ($\pm 1$ standard deviation) curves are also shown for both types of skies.

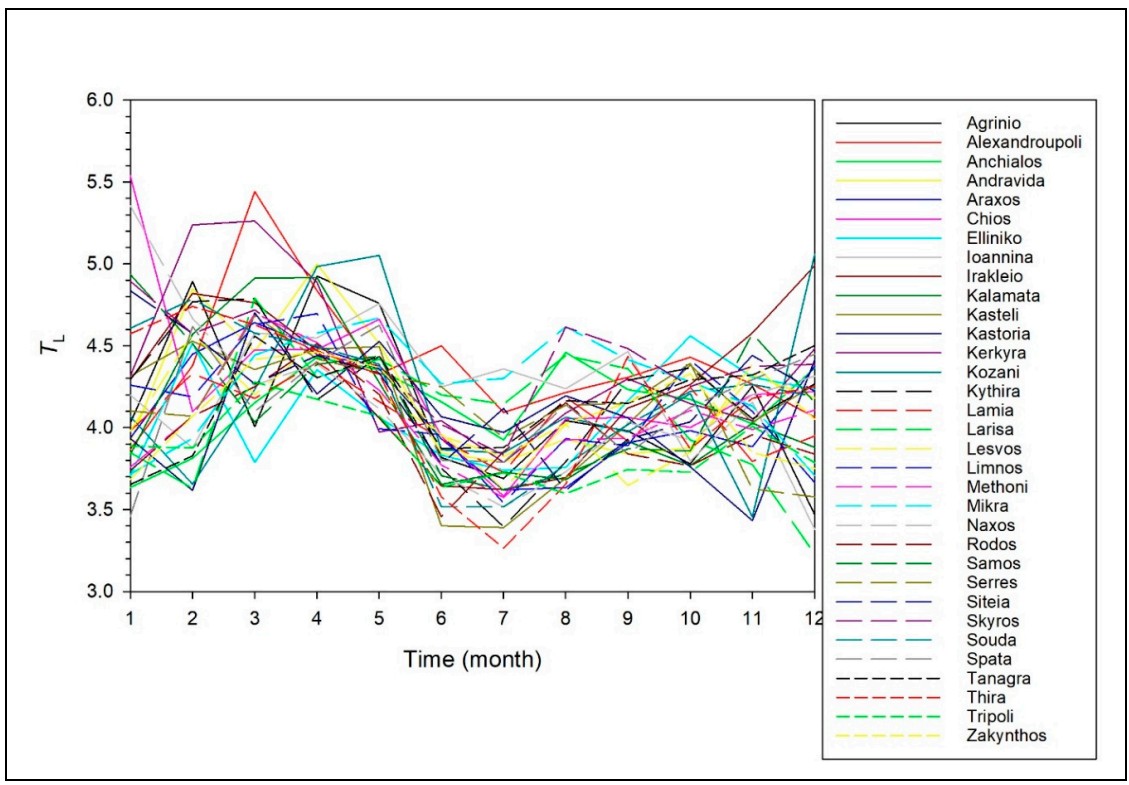

**Figure 10.** *Cont.*

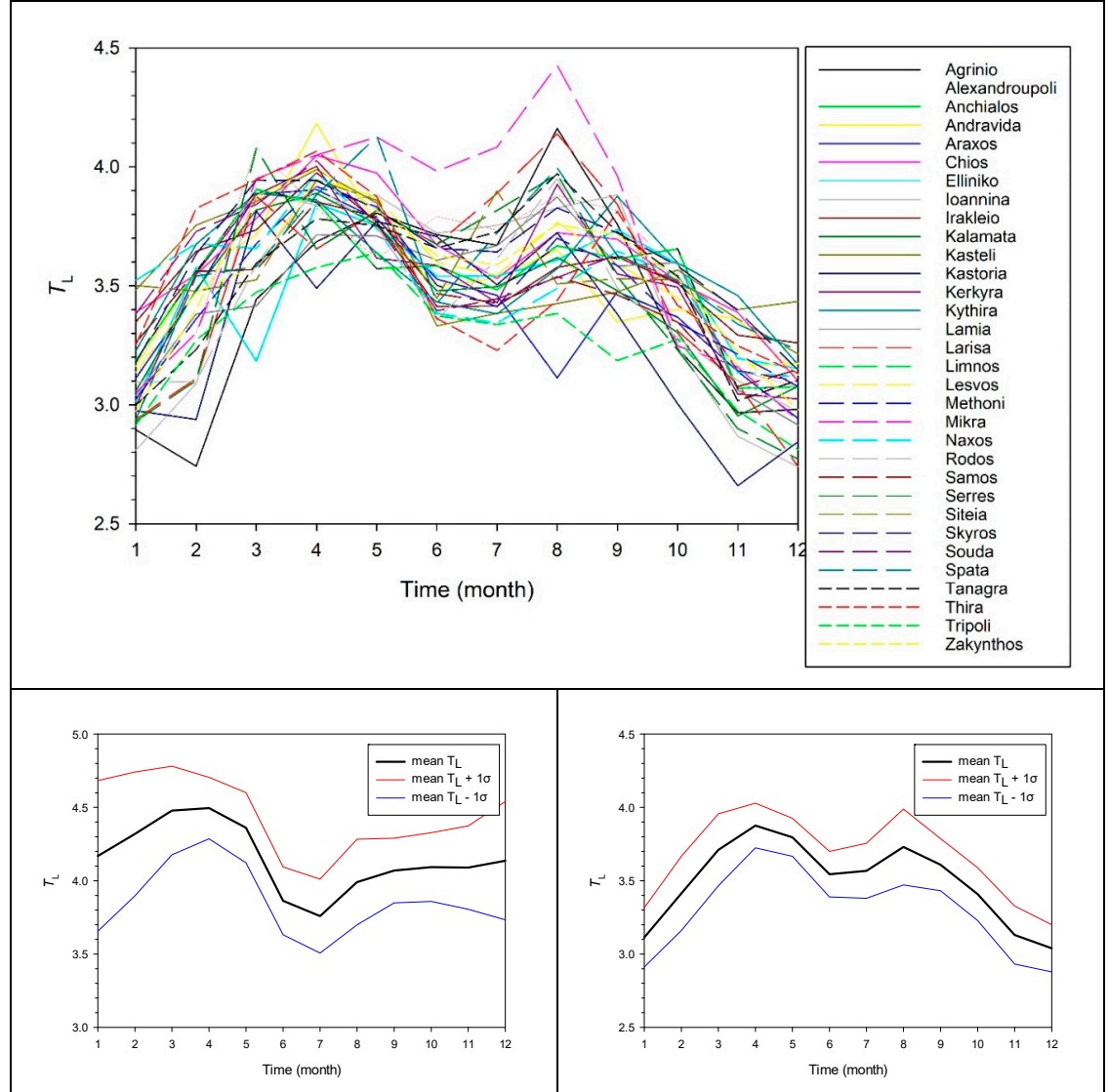

**Figure 10.** Intra-annual variation of the monthly mean $T_L$ values for all 33 stations during TATY. Upper graph: $T_L$ variation for all-sky conditions. Middle graph: $T_L$ variation for clear-sky conditions. Lower left graph: average $T_L$ variation for all-sky conditions (black line). Lower right graph: average $T_L$ variation for clear-sky conditions (black line). The red and blue lines represent the ± 1σ around the mean.

It is seen from Figure 10 that $T_L$ obtains a peak in March–April and shows a rather flat behavior after September for all-sky conditions (lower left graph). The black curve is the average of all 33 $T_L$ variations. Similar behavior is shown by the intra-annual variation of $T_L$ under clear skies. There are two pronounced peaks in April and August (lower right graph). Very similar intra-annual patterns and magnitudes of $T_L$ were reported by Pedrós et al. [72] for Valencia, Spain; by El-Wakil et al. [73] for Cairo, Egypt; by Diabate et al. [14] for several sites in Africa; and by Eftimie [74] (2009) for Braşov, Romania. The spring and autumn peaks may be due to the frequent wind streams from Northern Africa, which may bring desert dust over Greece. The Sahara-dust transport has been a rather frequent phenomenon in the recent 20 years [75]. On the other hand, the summer minimum may be attributed to the strong cleansing northeasterly winds (called *Etesians*), which make the atmosphere drier. Figure 11 is same as Figure 10, but it refers to $T_{UM}$. The peaks and the low summer values here have a similar interpretation to that given for $T_L$.

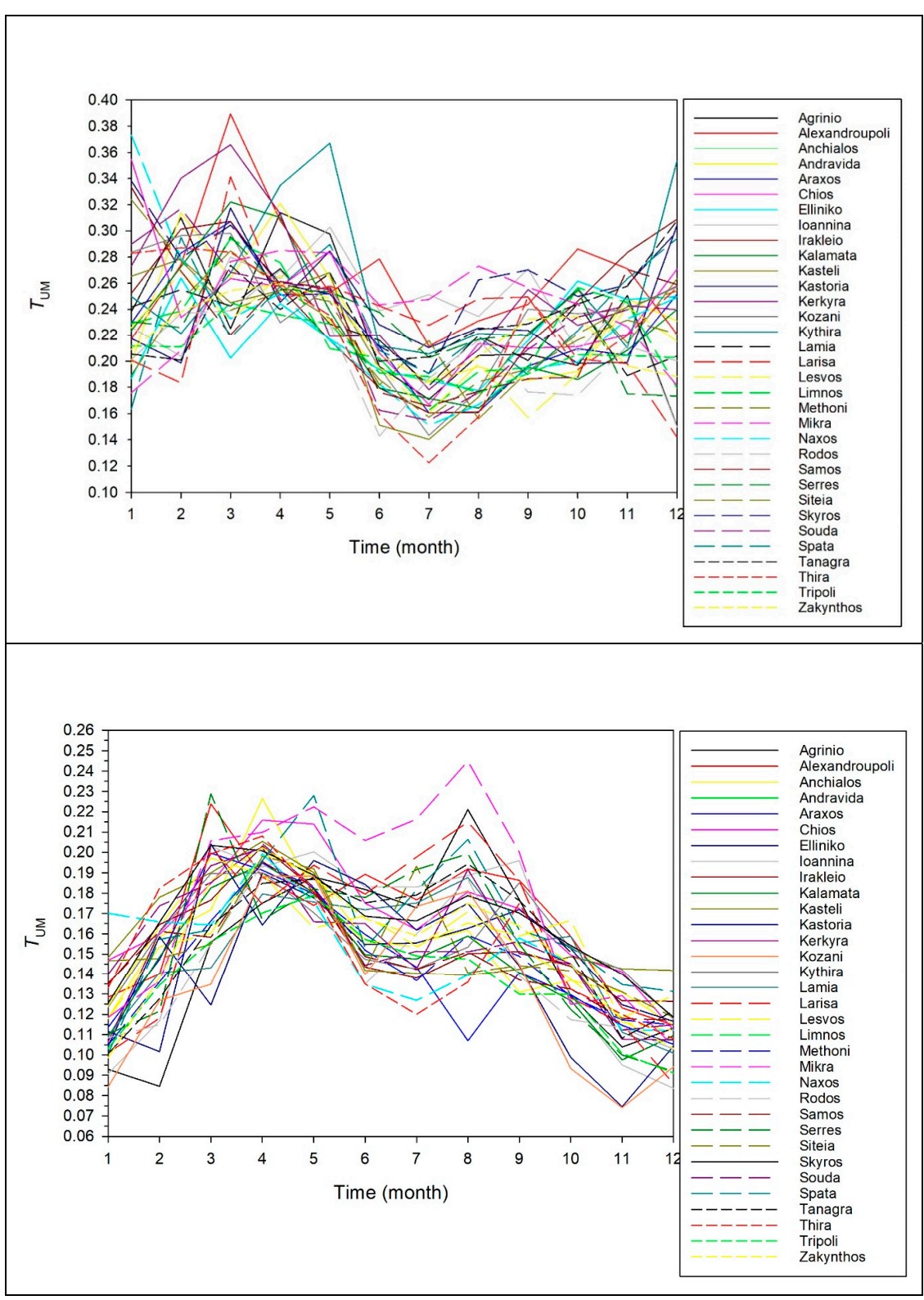

**Figure 11.** *Cont.*

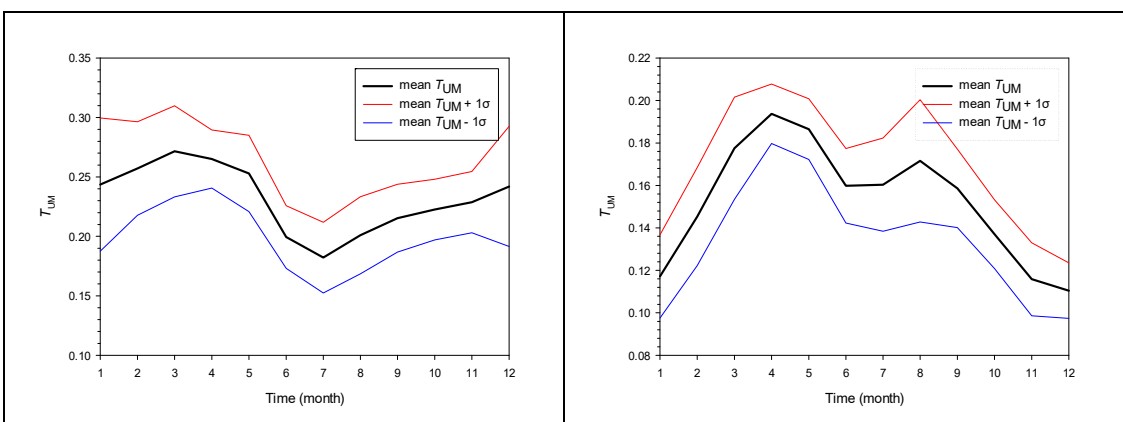

**Figure 11.** Intra-annual variation of the monthly mean $T_{UM}$ values for all 33 stations during TATY. Upper graph: $T_{UM}$ variation for all-sky conditions. Middle graph: $T_{UM}$ variation for clear-sky conditions. Lower left graph: average $T_{UM}$ variation for all-sky conditions (black line). Lower right graph: average $T_{UM}$ variation for clear-sky conditions (black line). The red and blue lines represent the $\pm 1\sigma$ around the mean.

### 3.5. Maps of Annual Mean $T_L$ and $T_{UM}$

Figures 12 and 13 present the distribution of the annual mean values of $T_L$ and $T_{UM}$ over the Greek territory, respectively, for all and clear-sky conditions. Two major peaks are seen in both $T_L$ and $T_{UM}$ values for all skies; one is centered over the South Ionian Sea and the other over the Central Aegean Sea. Both peaks are located in a northeast–southwest direction, and it is believed they are due to Sahara-dust transport mostly over Southern Greece in spring, late summer, and early autumn (first peak) and the turbulent air during high northeasterly winds over the Aegean Sea almost all-year round (second peak). During clear-sky conditions, the second peak is drastically reduced because it is related to the *Etesian* winds that blow during summertime and are associated with minimum humidity in the atmosphere.

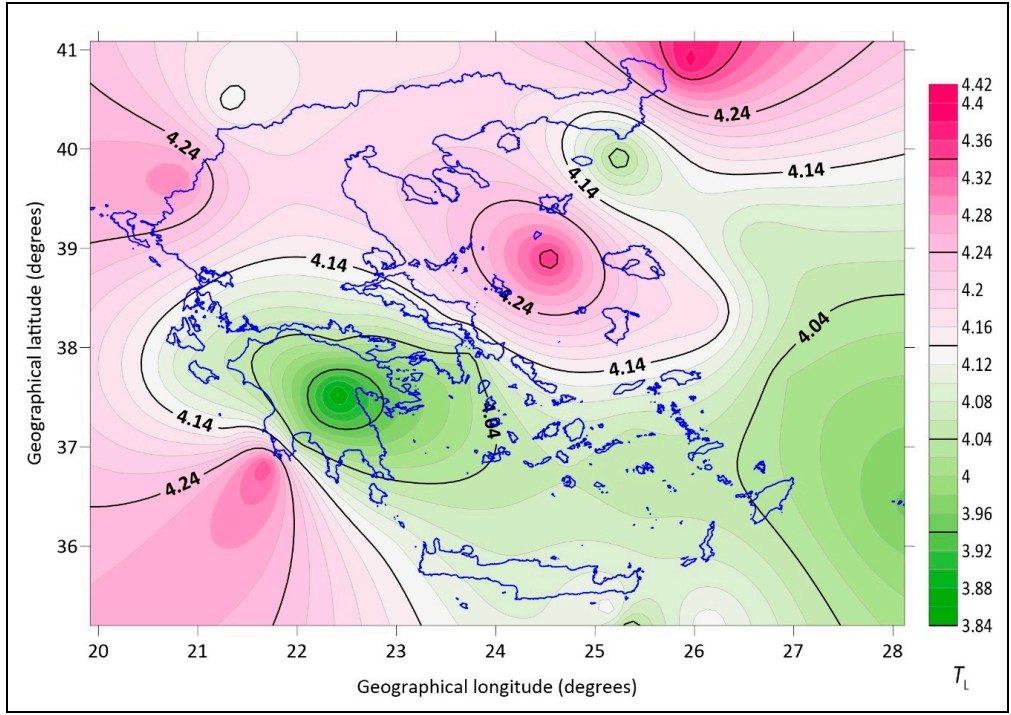

**Figure 12.** *Cont.*

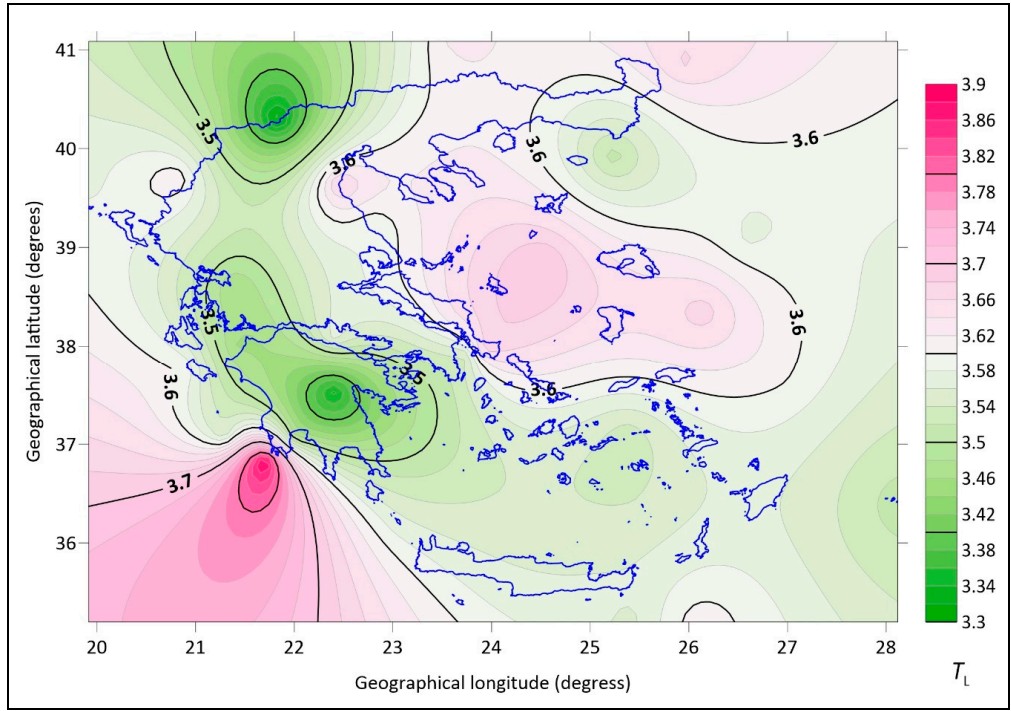

**Figure 12.** Map of annual mean $T_L$ values over Greece for all (upper graph) and clear-sky (lower graph) conditions during TATY.

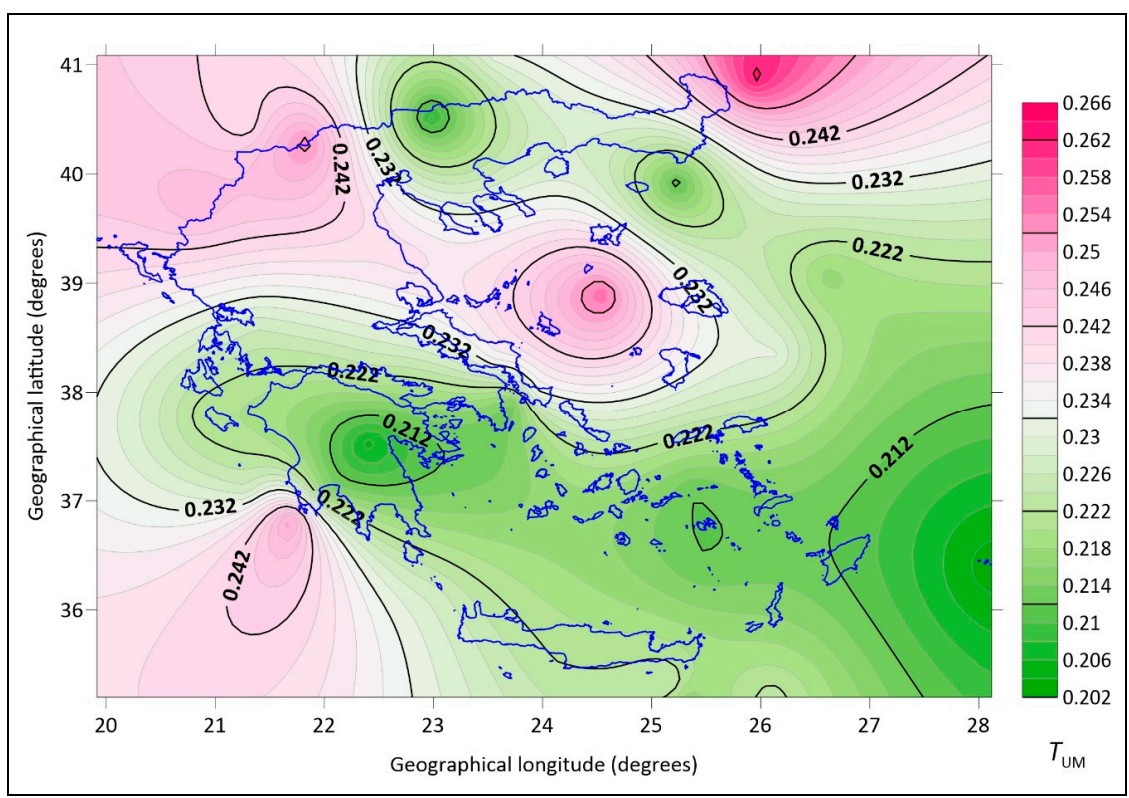

**Figure 13.** *Cont.*



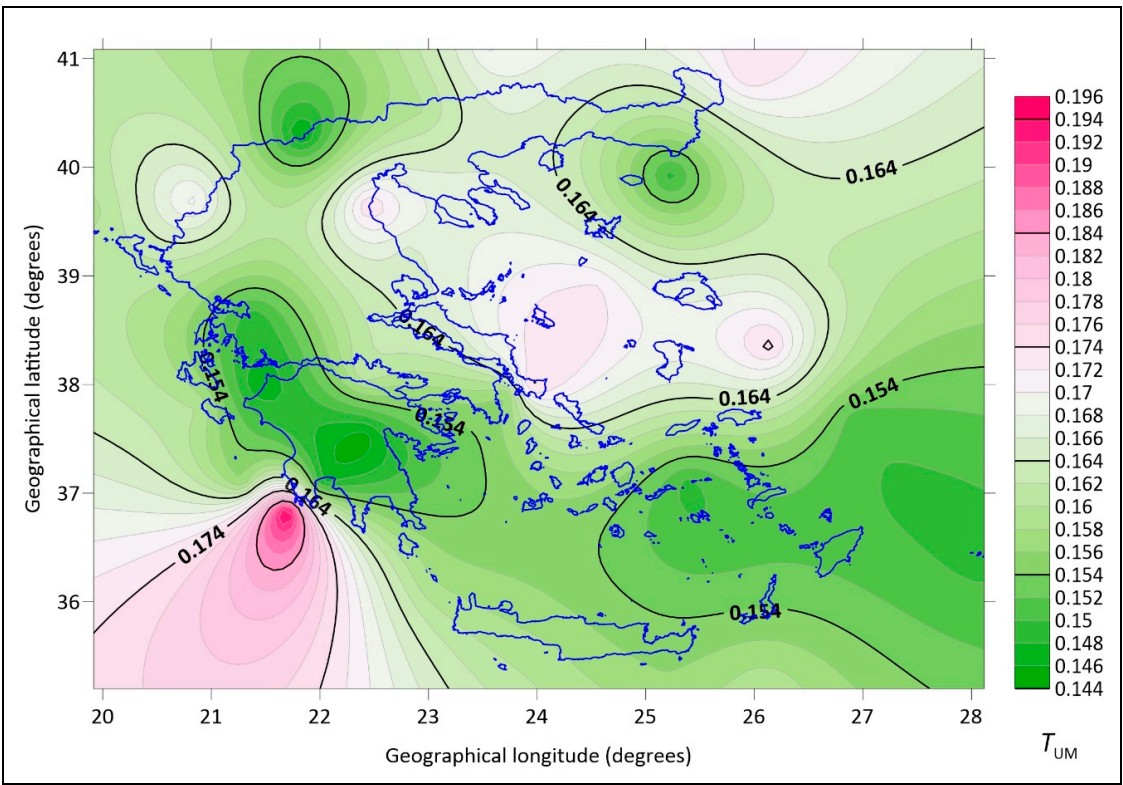

**Figure 13.** Map of annual mean $T_{UM}$ values over Greece for all (upper graph) and clear-sky (lower graph) conditions during TATY.

As far as the minima in the $T_L$ and $T_{UM}$ annual values are concerned, these are spotted over Peloponnese ($T_L$) or extended to the South Aegean Sea ($T_{UM}$). These broad minima may be thought as a dividing line between high-pressure cells over the Aegean and low-pressure cells in the Southern Ionian Sea.

### 3.6. Variation of $T_{UM}$ vs. $T_L$ over Greece

There are few studies worldwide that deal with the Unsworth–Monteith turbidity coefficient because of its dependence on the water-vapor content in the atmosphere, as stated in Section 3.1. However, those that exist show a straightforward linear dependence of $T_{UM}$ on $T_L$ and vice versa. Figure 14 shows this dependence for all and clear-sky conditions over Greece.

Kambezidis et al. [24] provided the relationship $T_{UM} = -0.11 + 0.07\ T_L$, $R^2 = 0.88$ for Athens in the period 1975–1984. Furthermore, Kambezidis et al. [25] provided the expression $T_{UMvis} = -0.1338 + 0.1360\ T_{Lvis}$, $R^2 = 0.99$ for Athens in the period 1992–1995, where *vis* refers to the visible band of the solar spectrum. In an updated work about the Linke and Unsworth–Montheith turbidity parameters for Athens, Kambezidis et al. [26] provided the relationship $T_{UM} = -0.1690 + 0.0939\ T_L$, $R^2 = 0.97$ in the period 1975–1995, while Jacovides et al. [76] derived the expression $T_{UM} = -0.182 + 0.0837\ T_L$, $R^2 = 0.99$ for the same site in the period 1954–1991. It is interesting to note that the coefficients of these equations for Athens are in close agreement with those provided for clear-sky conditions for all climatic zones in Greece (see Figure 14). Pedrós et al. [72] provided a relationship for $T_{UM}$ as a function of $\beta$ for Valencia, Spain in the period 1990–1996 instead of $T_L$.

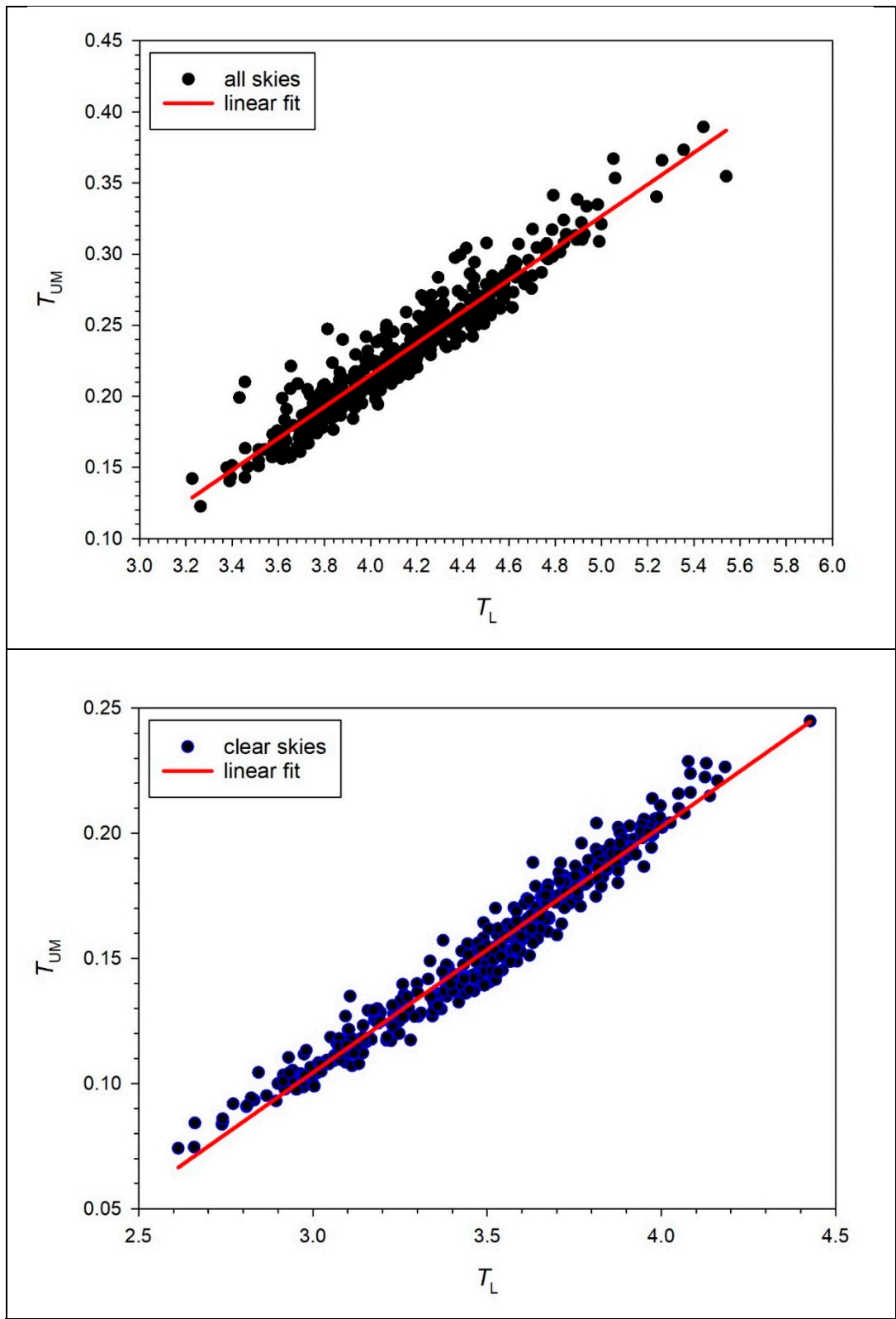

**Figure 14.** Monthly mean $T_{UM}$ vs. $T_L$ data pairs for all sites under all (upper graph) and clear-sky (lower graph) conditions in TATY and their best-fit lines. Linear regression equations: for all-skies $T_{UM}$ = −0.2320 + 0.1120 $T_L$, $R^2$ = 0.91; for clear skies $T_{UM}$ = −0.1900 + 0.0980 $T_L$, $R^2$ = 0.96.

## 4. Conclusions

The present study estimated the Linke ($T_L$) and Unsworth–Monteith ($T_{UM}$) turbidity parameters at 33 sites in Greece under all and clear-sky conditions over a year. That year was considered typical (Typical Atmospheric Turbidity Year, TATY) because the estimation of both turbidity factors was based on values of the meteorological parameters that comprise recently-derived Typical Meteorological Years for the same sites. Each TATY consists of the same TMMs as in the TMYs-PV for the specific location.

The present work presented some innovations: (i) the notion of TATY was introduced for the first time worldwide; (ii) turbidity-parameter maps for Greece were derived for the first time; (iii) the definition of the modified diffuse fraction was introduced for the first time worldwide; and (iv) use of the climatic zones of Greece for energy purposes was made, since atmospheric turbidity is used in advanced solar codes to estimate solar radiation at a location destined for energy applications (e.g., PV installations). For the sake of simplicity, two representative sites per climatic zone were selected in some parts of the analysis.

Month–hour graphs for $T_L$ and $T_{UM}$ were prepared for the eight selected sites. For all-sky conditions, relatively high values of $T_L$ were found in the mornings and evenings during January–March or even early April; lower values were found in the other months of the year. The $T_{UM}$ variation indicated a similar pattern to that of $T_L$ with a slight differentiation because of the dependence of $T_{UM}$ on the water vapor in the atmosphere. Under clear skies, the $T_L$ month–hour graphs showed a pattern resembling the all-sky pattern. The morning and evening peaks were retained in the period January–March, with maximum values in the morning hours throughout the year, but these were lower than those in late winter–early spring. In the evening (late spring–early winter), the $T_L$ values were lower than the morning values. $T_{UM}$ followed the $T_L$ pattern as in the case of all skies. Similar patterns were found for the other 25 sites, but they were not shown in the study for space-saving reasons.

The intra-annual variation of both parameters showed a pattern with maximum values in early spring and late summer and lower values in summertime and most winter months. This behavior was found to be consistent with the intra-annual variation of the turbidity factors at other sites in Europe and Africa.

Maps of the annual mean values of the turbidity parameters over Greece for all and clear-sky conditions were prepared. Persistent low values were found over Peloponnese all over the year and higher values in the southern part of the Ionian Sea.

Because of the difficulty in estimating $T_{UM}$ (i.e., more elaborate calculations than for $T_L$; compare methodologies in Sections 2.1.1 and 2.1.2), linear expressions with high coefficients of determination ($R^2$) were given for $T_{UM}$ as function of $T_L$ for all sites and both all and clear-sky conditions.

Finally, an attempt to facilitate the estimation of the diffuse horizontal solar radiation at any location in Greece was made, if the global horizontal radiation is known or can be estimated via a solar code. Linear expressions of $k'_d$ vs. $k'_t$ were derived for the sites in each climatic zone as well as for all sites regardless of zone. The expressions provided high values of $R^2$, thus showing their applicability.

**Author Contributions:** Conceptualisation, methodology, data collection, data analysis, and writing—original draft preparation, H.D.K.; methodology, mathematical formulation, review and editing, B.E.P. All authors have read and agreed to the published version of the manuscript.

**Funding:** This research was implemented in the frame of the KRIPIS-THESPIA-II project, grant number MIS 5002517, funded by the General Secretariat of Research and Technology in Greece.

**Acknowledgments:** The authors are thankful to HNMS for the disposition of the meteorological data from 33 stations of the network with the purpose of building the Typical Meteorological Years for Greece and, therefore, the Typical Atmospheric Turbidity Years in the present study.

**Conflicts of Interest:** The authors declare no conflict of interest.

## Nomenclature

*Greek symbols*

| | |
|---|---|
| $\alpha$ | Ångström exponent (dimensionless) |
| $\beta$ | Ångström turbidity coefficient (dimensionless) |
| $\gamma$ | Solar elevation or solar height or solar altitude (degrees) |
| $\lambda$ | Geographical longitude (degrees, positive in east of Greenwich) |
| $\varphi$ | Geographical latitude (degrees, positive in Northern Hemisphere) |

*Latin symbols*

| | |
|---|---|
| $B$ | Schüepp turbidity factor (dimensionless) |
| $B_e$ | Direct horizontal irradiance ($\mathrm{Wm^{-2}}$) |
| $B^*_e$ | Direct (normal) irradiance in a dust-free atmosphere ($\mathrm{Wm^{-2}}$) |
| $D$ | Day of the year (dimensionless); $D = 1$ for 1 January, 365 for 31 December in a non-leap year and 366 in a leap year |
| $D_e$ | Diffuse horizontal irradiance ($\mathrm{Wm^{-2}}$) |
| $e_m$ | Partial water-vapor pressure (hpa) |
| $e_s$ | Saturation water-vapor pressure (hpa) |
| $G_e$ | Global horizontal irradiance ($\mathrm{Wm^{-2}}$) |
| $G_{e,extra}$ | Extra-terrestrial solar irradiance ($\mathrm{Wm^{-2}}$) = $S\,G_{e,o}$ |
| $G_{e,o}$ | Solar constant = 1361.1 $\mathrm{Wm^{-2}}$ |
| $m$ | Optical air mass (dimensionless) |
| $m'$ | Pressure-corrected optical air mass (dimensionless) |
| $\overline{k_r}$ | Mean attenuation of the direct solar radiation due to Rayleigh scattering (dimensionless) |
| $k_d$ | Diffuse fraction (dimensionless) |
| $k'_d$ | Modified diffuse fraction (dimensionless) |
| $k_t$ | Clearness index (dimensionless) |
| $k'_t$ | Modified clearness index (dimensionless) |
| $P_o$ | Sea-level atmospheric pressure = 1013.25 hpa |
| $P_z$ | Atmospheric pressure at height z (hpa) |
| $RH$ | Relative humidity (%) |
| $S$ | Correction of the Earth–Sun distance (dimensionless) |
| $T_i$ | Atmospheric transmittance due to $i^{th}$ atmospheric constituent (dimensionless) |
| $T_L$ | Linke turbidity factor (dimensionless) |
| $t$ | Ambient temperature (K) = 273.15 + t' |
| $t'$ | Ambient temperature (°C) |
| $T_{UM}$ | Unsworth–Monteith turbidity coefficient (dimensionless) |
| $u_i$ | Atmospheric column content due to $i^{th}$ atmospheric constituent (atm-cm, or cm, depending on the constituent) |
| $z$ | Altitude (m) |

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
