# Peer review of "Climatology of the Linke and Unsworth–Monteith Turbidity Parameters for Greece: Introduction to the Notion of a Typical Atmospheric Turbidity Year"

_applsci, doi:10.3390/app10114043_

Round 1

Reviewer 1 Report

Dear authors,

you did a good job in the description of an average year looking atmosphere turbidity.

However I found it hard to understand your very first equation because you put it in line 70 and the its expalnation starts at line 74. I think you should re-order explanations and equations. Nomenclature is not obvious for everybody.

When you expose the state of art, your last citations are from 2016-2017. Are you shure that nothing has been said lastly?  for example:

https://www.sciencedirect.com/science/article/pii/S0169809519309792

Speaking about references, I have not found your reference number 28 cited in your paper.

In table 2 you put the list of your stations. Ithink tha it should hav a column stating the climatic region, it would be clarifying.

In figure 3 numbers of stations are quite illegible for me. In its foot you say that information is taken from your reference TOTEE (60) but in your references list TOTEE is number 61.

In figures 4 to 7 you put a colour code, bt I cannot read it. In the figure it is on a black rectangle, and nothing written in it.

Author Response

We thank Reviewer 1 for his/her constructive comments. Almost all of them have been incorporated in the revised version of the manuscript. More specifically, our replies to his/her queries are as follows.

Query 1. However I found it hard to understand your very first equation because you put it in line 70 and the its expalnation starts at line 74. I think you should re-order explanations and equations. Nomenclature is not obvious for everybody.

Reply 1. Re-ordering of Equations in Section 2.1.1 has taken place; a Nomenclature has been added in the beggining of the article.

Query 2. When you expose the state of art, your last citations are from 2016-2017. Are you shure that nothing has been said lastly?  for example:

https://www.sciencedirect.com/science/article/pii/S0169809519309792.

Reply 2. The reference has been added.

Query 3. Speaking about references, I have not found your reference number 28 cited in your paper.

Reply 3. It existed just above Equation (2) in the original version and now it has been placed just after Equation (1a) in the revised version.

Query 4. In table 2 you put the list of your stations. I think tha it should have a column stating the climatic region, it would be clarifying.

Reply 4. A new column with the climatic zones of the stations has been added in Table 2.

Query 5. In figure 3 numbers of stations are quite illegible for me. In its foot you say that information is taken from your reference TOTEE (60) but in your references list TOTEE is number 61.

Reply 5. The map has now been enlarged so that the numbers of the stations be legible. The reference of TOTEE has been changed from 60 to 61.

Query 6. In figures 4 to 7 you put a colour code, but I cannot read it. In the figure it is on a black rectangle, and nothing written in it.

Reply 6. We are sorry, but the colour codes do exist and are legible. Probably there is some misfunction in your PC and you could not see them properly. One could see them clearly upon downloading the manuscript from the journal’s website.

Reviewer 2 Report

The paper studies the Linke turbidity factor and the Unsworth-Monteith turbidity coefficient. The required data measured in 33 meteorological stations located in Greece were adopted for estimating the turbidities. It is an interesting and well-written paper and there are a few minor comments as follows.

Page 3, Line 98: Whether Be* should be direct normal solar radiation.

There are many climatic parameters for the calculations and nomenclature is required.

Figures 1, 8, 9, 10, 11 and 14: Use point instead of comma to indicate the decimal.

Page 4, Line 126: What is ‘D’ in Equation 9g?

Figure 11b: The legend is quite vague.

Author Response

We thank Reviewer 2 for his/her constructive comments. All of them have been incorporated in the revised version of the manuscript. More specifically, our replies to his/her queries are as follows.

Query 1. Page 3, Line 98: Whether Be* should be direct normal solar radiation.

Reply 1. It is now indicated in the nomenclature that it refers to the direct-normal irradiance.

Query 2. There are many climatic parameters for the calculations and Nomenclature is required.

Reply 2. A Nomenclature has now been added in the beginning of the article explaining all variables used in the paper.

Query 3. Figures 1, 8, 9, 10, 11 and 14: Use point instead of comma to indicate the decimal.

Reply 3. Figs. 1, 8, 9, 10, 11, 12, 13, and 14 have been re-drawn; the decimal separators are now dots and the legends are clearer.

Query 4. Page 4, Line 126: What is ‘D’ in Equation 9g?

Reply 4. D is the day of the year, also indicated in the Nomenclature.

Query 5. Figure 11b: The legend is quite vague.

Reply 5. The figure has been re-drawn and there is more clarity now.